# FEDERATED NEURAL BANDITS

**Zhongxiang Dai, Yao Shu,**\* **Arun Verma, Flint Xiaofeng Fan & Bryan Kian Hsiang Low**
Department of Computer Science, National University of Singapore
`{daizhongxiang,shuyao,arun,xiaofeng,lowkh}@comp.nus.edu.sg`

**Patrick Jaillet**
Department of Electrical Engineering and Computer Science, MIT
`jaillet@mit.edu`

## ABSTRACT

Recent works on *neural contextual bandits* have achieved compelling performances due to their ability to leverage the strong representation power of neural networks (NNs) for reward prediction. Many applications of contextual bandits involve multiple agents who collaborate without sharing raw observations, thus giving rise to the setting of *federated contextual bandits*. Existing works on federated contextual bandits rely on linear or kernelized bandits, which may fall short when modeling complex real-world reward functions. So, this paper introduces the *federated neural-upper confidence bound* (FN-UCB) algorithm. To better exploit the federated setting, FN-UCB adopts a weighted combination of two UCBs: $\text{UCB}^a$ allows every agent to additionally use the observations from the other agents to accelerate exploration (without sharing raw observations), while $\text{UCB}^b$ uses an NN with aggregated parameters for reward prediction in a similar way to federated averaging for supervised learning. Notably, the weight between the two UCBs required by our theoretical analysis is amenable to an interesting interpretation, which emphasizes $\text{UCB}^a$ initially for *accelerated exploration* and relies more on $\text{UCB}^b$ later after enough observations have been collected to train the NNs for accurate reward prediction (i.e., *reliable exploitation*). We prove sub-linear upper bounds on both the cumulative regret and the number of communication rounds of FN-UCB, and empirically demonstrate its competitive performance.

## 1 INTRODUCTION

The stochastic multi-armed bandit is a prominent method for sequential decision-making problems due to its principled ability to handle the exploration-exploitation trade-off (Auer, 2002; Bubeck & Cesa-Bianchi, 2012; Lattimore & Szepesvári, 2020). In particular, the stochastic contextual bandit problem has received enormous attention due to its widespread real-world applications such as recommender systems (Li et al., 2010a), advertising (Li et al., 2010b), and healthcare (Greenewald et al., 2017). In each iteration of a stochastic contextual bandit problem, an agent receives a context (i.e., a $d$-dimensional feature vector) for each of the $K$ arms, selects one of the $K$ contexts/arms, and observes the corresponding reward. The goal of the agent is to sequentially pull the arms in order to maximize the cumulative reward (or equivalently, minimize the *cumulative regret*) in $T$ iterations.

To minimize the cumulative regret, *linear contextual bandit* algorithms assume that the rewards can be modeled as a linear function of the input contexts (Dani et al., 2008) and select the arms via classic methods such as upper confidence bound (UCB) (Auer, 2002) or Thompson sampling (TS) (Thompson, 1933), consequently yielding the Linear UCB (Abbasi-Yadkori et al., 2011) and Linear TS (Agrawal & Goyal, 2013) algorithms. The potentially restrictive assumption of a linear model was later relaxed by *kernelized contextual bandit* algorithms (Chowdhury & Gopalan, 2017; Valko et al., 2013), which assume that the reward function belongs to a reproducing kernel Hilbert space (RKHS) and hence model the reward function using kernel ridge regression or Gaussian process (GP) regression. However, this assumption may still be restrictive (Zhou et al., 2020) and the kernelized

---

\*Corresponding author.

model may fall short when the reward function is very complex and difficult to model. To this end, neural networks (NNs), which excel at modeling complex real-world functions, have been adopted to model the reward function in contextual bandits, thereby leading to *neural contextual bandit* algorithms such as Neural UCB (Zhou et al., 2020) and Neural TS (Zhang et al., 2021). Due to their ability to use the highly expressive NNs for better reward prediction (i.e., *exploitation*), Neural UCB and Neural TS have been shown to outperform both linear and kernelized contextual bandit algorithms in practice. Moreover, the cumulative regrets of Neural UCB and Neural TS have been analyzed by leveraging the theory of the *neural tangent kernel* (NTK) (Jacot et al., 2018), hence making these algorithms both provably efficient and practically effective. We give a comprehensive review of the related works on neural bandits in App. A.

The contextual bandit algorithms discussed above are only applicable to problems with a single agent. However, many modern applications of contextual bandits involve multiple agents who (*a*) collaborate with each other for better performances and yet (*b*) are unwilling to share their raw observations (i.e., the contexts and rewards). For example, companies may collaborate to improve their contextual bandits-based recommendation algorithms without sharing their sensitive user data (Huang et al., 2021b), while hospitals deploying contextual bandits for personalized treatment may collaborate to improve their treatment strategies without sharing their sensitive patient information (Dai et al., 2020). These applications naturally fall under the setting of *federated learning* (FL) (Kairouz et al., 2019; Li et al., 2021) which facilitates collaborative learning of supervised learning models (e.g., NNs) without sharing the raw data. In this regard, a number of *federated contextual bandit* algorithms have been developed to allow bandit agents to collaborate in the federated setting (Shi & Shen, 2021). We present a thorough discussion of the related works on federated contextual bandits in App. A. Notably, Wang et al. (2020) have adopted the Linear UCB policy and developed a mechanism to allow every agent to additionally use the observations from the other agents to **accelerate exploration**, while only requiring the agents to exchange some sufficient statistics instead of their raw observations. However, these previous works have only relied on either linear (Dubey & Pentland, 2020; Huang et al., 2021b) or kernelized (Dai et al., 2020; 2021) methods which, as discussed above, may lack the expressive power to model complex real-world reward functions (Zhou et al., 2020). Therefore, this naturally brings up the need to **use NNs for better exploitation** (i.e., reward prediction) in federated contextual bandits, thereby motivating the need for a *federated neural contextual bandit* algorithm.

To develop a federated neural contextual bandit algorithm, an important technical challenge is how to leverage the federated setting to simultaneously (*a*) **accelerate exploration** by allowing every agent to additionally use the observations from the other agents without requiring the exchange of raw observations (in a similar way to that of Wang et al. (2020)), and (*b*) **improve exploitation** by further enhancing the quality of the NN for reward prediction through the federated setting (i.e., without requiring centralized training using the observations from all agents). In this work, we provide a theoretically grounded solution to tackle this challenge by deploying a weighted combination of two upper confidence bounds (UCBs). The first UCB, denoted by $\text{UCB}^a$, incorporates the neural tangent features (i.e., the random features embedding of NTK) into the Linear UCB-based mechanism adopted by Wang et al. (2020), which achieves the first goal of accelerating *exploration*. The second UCB, denoted by $\text{UCB}^b$, adopts an aggregated NN whose parameters are the average of the parameters of the NNs trained by all agents using their local observations for better reward prediction (i.e., better *exploitation* in the second goal). Hence, $\text{UCB}^b$ improves the quality of the NN for reward prediction in a similar way to the most classic FL method of federated averaging (FedAvg) for supervised learning (McMahan et al., 2017). Notably, our choice of the weight between the two UCBs, which naturally arises during our theoretical analysis, has an interesting practical interpretation (Sec. 3.3): More weight is given to $\text{UCB}^a$ in earlier iterations, which allows us to use the observations from the other agents to accelerate the exploration in the early stage; more weight is assigned to $\text{UCB}^b$ only in later iterations after every agent has collected enough local observations to train its NN for accurate reward prediction (i.e., reliable exploitation). Of note, our novel design of the weight (Sec. 3.3) is crucial for our theoretical analysis and may be of broader interest for future works on related topics.

This paper introduces the first federated neural contextual bandit algorithm which we call *federated neural-UCB* (`FN-UCB`) (Sec. 3). We derive an upper bound on its total cumulative regret from all $N$ agents: $R_T = \widetilde{O}(\widetilde{d}\sqrt{TN} + \widetilde{d}_{\max} N\sqrt{T})^1$ where $\widetilde{d}$ is the effective dimension of the contexts from all $N$ agents and $\widetilde{d}_{\max}$ represents the maximum among the $N$ individual effective dimensions

---

[1] The $\widetilde{\mathcal{O}}$ ignores all logarithmic factors.

of the contexts from the $N$ agents (Sec. 2). The communication complexity (i.e., total number of communication rounds in $T$ iterations) of FN-UCB can be upper-bounded by $C_T = \widetilde{\mathcal{O}}(\widetilde{d}\sqrt{N})$. Finally, we use both synthetic and real-world contextual bandit experiments to explore the interesting insights about our FN-UCB and demonstrate its effective practical performance (Sec. 5).

## 2 BACKGROUND AND PROBLEM SETTING

Let $[k]$ denote the set $\{1, 2, \ldots, k\}$ for a positive integer $k$, $\mathbf{0}_k$ represent a $k$-dimensional vector of 0's, and $\mathbf{0}_{k \times k}$ denote an all-zero matrix with dimension $k \times k$. Our setting involves $N$ agents with the same reward function $h$ defined on a domain $\mathcal{X} \subset \mathbb{R}^d$. We consider centralized and synchronous communication: The communication is coordinated by a central server and every agent exchanges information with the central server during a *communication round*. In each iteration $t \in [T]$, every agent $i \in [N]$ receives a set $\mathcal{X}_{t,i} \triangleq \{x_{t,i}^k\}_{k \in [K]}$ of $K$ context vectors and selects one of them $x_{t,i} \in \mathcal{X}_{t,i}$ to be queried to observe a noisy output $y_{t,i} \triangleq h(x_{t,i}) + \epsilon$ where $\epsilon$ is an $R$-sub-Gaussian noise. We will analyze the total *cumulative regret* from all $N$ agents in $T$ iterations: $R_T \triangleq \sum_{i=1}^N \sum_{t=1}^T r_{t,i}$ where $r_{t,i} \triangleq h(x_{t,i}^*) - h(x_{t,i})$ and $x_{t,i}^* \triangleq \arg\max_{x \in \mathcal{X}_{t,i}} h(x)$.

Let $f(x; \theta)$ denote the output of a fully connected NN for input $x$ with parameters $\theta$ (of dimension $p_0$) and $g(x; \theta)$ denote the corresponding (column) gradient vector. We focus on NNs with ReLU activations, and use $L \geq 2$ and $m$ to denote its depth and width (of every layer), respectively. We follow the initialization technique from Zhang et al. (2021); Zhou et al. (2020) to initialize the NN parameters $\theta_0 \sim \text{init}(\cdot)$. Of note, as a common ground for collaboration, we let all $N$ agents share the same initial parameters $\theta_0$ when training their NNs and computing their *neural tangent features*: $g(x; \theta_0)/\sqrt{m}$ (i.e., the random features embedding of NTK (Zhang et al., 2021)). Also, let $\bar{\mathbf{H}}$ denote the $(TKN) \times (TKN)$-dimensional NTK matrix on the set of all received $TKN$ contexts (Zhang et al., 2021; Zhou et al., 2020). Similarly, let $\mathbf{H}_i$ denote the $(TK) \times (TK)$-dimensional NTK matrix on the set of $TK$ contexts received by agent $i$. We defer the details on the definitions of $\mathbf{H}$ and $\mathbf{H}_i$'s, the NN $f(x; \theta)$, and the initialization scheme $\theta_0 \sim \text{init}(\cdot)$ to App. B due to limited space.

Next, let $\mathbf{h} \triangleq [h(x_{t,i}^k)]_{t \in [T], i \in [N], k \in [K]}$ denote the $(TKN)$-dimensional column vector of reward function values at all received contexts and $B$ be an absolute constant s.t. $\sqrt{2\mathbf{h}^\top \mathbf{H}^{-1} \mathbf{h}} \leq B$. This is related to the commonly adopted assumption in kernelized bandits that $h$ lies in the RKHS $\mathcal{H}$ induced by the NTK (Chowdhury & Gopalan, 2017; Srinivas et al., 2010) (or, equivalently, that the RKHS norm $\|h\|_{\mathcal{H}}$ of $h$ is upper-bounded by a constant because $\sqrt{\mathbf{h}^\top \mathbf{H}^{-1} \mathbf{h}} \leq \|h\|_{\mathcal{H}}$ (Zhou et al., 2020). Following the works of Zhang et al. (2021); Zhou et al. (2020), we define the effective dimension of $\mathbf{H}$ as $\widetilde{d} \triangleq \frac{\log \det(I + \mathbf{H}/\lambda)}{\log(1 + TKN/\lambda)}$ with regularization parameter $\lambda > 0$. Similarly, we define the effective dimension for agent $i$ as $\widetilde{d}_i \triangleq \frac{\log \det(I + \mathbf{H}_i/\lambda)}{\log(1 + TK/\lambda)}$ and also define $\widetilde{d}_{\max} \triangleq \max_{i \in [N]} \widetilde{d}_i$. Note that the effective dimension is related to the *maximum information gain* $\gamma$ which is a commonly adopted notion in kernelized bandits (Zhang et al., 2021): $\widetilde{d} \leq 2\gamma_{TKN}/\log(1 + TKN/\lambda)$ and $\widetilde{d}_i \leq 2\gamma_{TK}/\log(1 + TK/\lambda), \forall i \in [N]$. Consistent with the works on neural contextual bandits (Zhang et al., 2021; Zhou et al., 2020), our only assumption on the reward function $h$ is its boundedness: $|h(x)| \leq 1, \forall x \in \mathcal{X}$. We also make the following assumptions for our theoretical analysis, all of which are mild and easily achievable, as discussed in (Zhang et al., 2021; Zhou et al., 2020):

**Assumption 1.** *There exists $\lambda_0 > 0$ s.t. $\mathbf{H} \succeq \lambda_0 I$ and $\mathbf{H}_i \succeq \lambda_0 I, \forall i \in [N]$. Also, all contexts satisfy $\|x\|_2 = 1$ and $[x]_j = [x]_{j+d/2}, \forall x \in \mathcal{X}_{t,i}, \forall t \in [T], i \in [N]$.*

## 3 FEDERATED NEURAL-UPPER CONFIDENCE BOUND (FN-UCB)

Our FN-UCB algorithm is described in Algo. 1 (agents' part) and Algo. 2 (central server's part).

### 3.1 OVERVIEW OF FN-UCB ALGORITHM

Before the beginning of the algorithm, we sample the initial parameters $\theta_0$ and share it with all agents (Sec. 2). In each iteration $t \in [T]$, every agent $i \in [N]$ receives a set $\mathcal{X}_{t,i} = \{x_{t,i}^k\}_{k \in [K]}$ of $K$ contexts (line 3 of Algo. 1) and then uses a weighted combination of $\text{UCB}_{t,i}^a$ and $\text{UCB}_{t,i}^b$ to select a context $x_{t,i} \in \mathcal{X}_{t,i}$ to be queried (lines 4-7 of Algo. 1). Next, each agent $i$ observes a noisy output $y_{t,i}$ (line 8 of Algo. 1) and then updates its local information (lines 9-10 of Algo. 1). After that, every agent checks if it has collected enough information since the last communication round (i.e., checks

---

**Algorithm 1** FN-UCB (Agent $i$)

---

1: **inputs:** $\lambda = 1 + 2/T$, $\theta_0 \sim \text{init}(\cdot)$, $W_{\text{sync}} = \mathbf{0}_{p_0 \times p_0}$, $W_{\text{new},i} = \mathbf{0}_{p_0 \times p_0}$, $B_{\text{sync}} = \mathbf{0}$, $B_{\text{new},i} = \mathbf{0}_{p_0}$, $\alpha = 0$, $V_{t,i}^{\text{local}} = \lambda I$, $V_{\text{sync,NN}}^{-1} = \lambda^{-1} I$, $V_{\text{last}} = \lambda I$, $t_{\text{last}} = 0$, $\theta_{\text{sync,NN}} = \theta_0$.
2: **for** $t = 1, 2, \ldots, T$ **do**
3:      Receive a set $\mathcal{X}_{t,i} = \{x_{t,i}^k\}_{k \in [K]}$ of $K$ contexts
4:      Compute $\overline{V}_{t,i} = \lambda I + W_{\text{sync}} + W_{\text{new},i}$ , $\overline{\theta}_{t,i} = \overline{V}_{t,i}^{-1}(B_{\text{sync}} + B_{\text{new},i})$
5:      Compute $\text{UCB}_{t,i}^a(x) \triangleq \langle g(x;\theta_0)/\sqrt{m}, \overline{\theta}_{t,i} \rangle + \nu_{TKN}\sqrt{\lambda} \left\| g(x;\theta_0)/\sqrt{m} \right\|_{\overline{V}_{t,i}^{-1}}$
6:      If $\alpha \neq 0$, compute $\text{UCB}_{t,i}^b(x) \triangleq f(x;\theta_{\text{sync,NN}}) + \nu_{TK}\sqrt{\lambda} N^{-1} \sum_{j=1}^N \left\| g(x;\theta_0)/\sqrt{m} \right\|_{(V_j^{\text{local}})^{-1}}$
7:      Select $x_{t,i} \triangleq \arg\max_{x \in \mathcal{X}_{t,i}} (1 - \alpha)\, \text{UCB}_{t,i}^a(x) + \alpha\, \text{UCB}_{t,i}^b(x)$
8:      Query $x_{t,i}$ to observe $y_{t,i}$
9:      Update $W_{\text{new},i} \leftarrow W_{\text{new},i} + g(x_{t,i};\theta_0)\, g(x_{t,i};\theta_0)^\top/m$, $B_{\text{new},i} \leftarrow B_{\text{new},i} + y_{t,i}\, g(x_{t,i};\theta_0)/\sqrt{m}$
10:      Update $V_{t,i}^{\text{local}} = V_{t-1,i}^{\text{local}} + g(x_{t,i};\theta_0)\, g(x_{t,i};\theta_0)^\top/m$
11:      **if** $(t - t_{\text{last}}) \log\left( \det(\lambda I + W_{\text{sync}} + W_{\text{new},i})/\det(V_{\text{last}}) \right) > D$ **then**
12:          Send a synchronisation signal to the central server to start a communication round
13:      **if** a communication round is started **then**
14:          Train an NN with gradient descent using all agent $i$'s *local* observations $\mathcal{D}_{t,i} = \{(x_{\tau,i}, y_{\tau,i})\}_{\tau \in [t]}$ based on initial parameters $\theta_0$, learning rate $\eta$, number $J$ of iterations, and Equation 1 as loss function to obtain $\theta_t^i$
15:          Compute $\alpha_{t,i} = \widetilde{\sigma}_{t,i,\min}^{\text{local}}/\widetilde{\sigma}_{t,i,\max}^{\text{local}}$ (Sec. 3.3)
16:      **send** $\{ W_{\text{new},i},\ B_{\text{new},i},\ \theta_t^i,\ \alpha_{t,i},\ (V_{t,i}^{\text{local}})^{-1} \}$ to the central server
17:      **receive** $\{ W_{\text{sync}},\ B_{\text{sync}},\ \theta_{\text{sync,NN}},\ \alpha,\ \{(V_i^{\text{local}})^{-1}\}_{i \in [N]} \}$ from the central server
18:      Set $V_{\text{last}} = W_{\text{sync}} + \lambda I$ , $t_{\text{last}} = t$ , $W_{\text{new},i} = \mathbf{0}_{p_0 \times p_0}$ , $B_{\text{new},i} = \mathbf{0}$

---

**Algorithm 2** Central Server

---

1: **if** a synchronization signal is received from *any agent* **then**
2:      Send a signal to all agents to start a communication round
3: **receive** $\{ W_{\text{new},i},\ B_{\text{new},i},\ \theta_t^i,\ \alpha_{t,i},\ (V_{t,i}^{\text{local}})^{-1} \}_{i \in [N]}$
4: Compute $\theta_{\text{sync,NN}} = N^{-1} \sum_{i=1}^N \theta_t^i$ , $\alpha = \min_{i \in [N]} \alpha_{t,i}$ ; let $(V_i^{\text{local}})^{-1} = (V_{t,i}^{\text{local}})^{-1}, \forall i \in [N]$
5: Update $W_{\text{sync}} \leftarrow W_{\text{sync}} + \sum_{i=1}^N W_{\text{new},i}$ , $B_{\text{sync}} \leftarrow B_{\text{sync}} + \sum_{i=1}^N B_{\text{new},i}$
6: Broadcast $\{ W_{\text{sync}},\ B_{\text{sync}},\ \theta_{\text{sync,NN}},\ \alpha,\ \{(V_i^{\text{local}})^{-1}\}_{i \in [N]} \}$ to all agents

---

the criterion in line 11 of Algo. 1); if so, it sends a synchronization signal to the central server (line 12 of Algo. 1) who then tells all agents to start a communication round (line 2 of Algo. 2). During a communication round, every agent $i$ uses its current history of local observations to train an NN (line 14 of Algo. 1) and sends its updated local information to the central server (line 16 of Algo. 1); the central server then aggregates these information from all agents (lines 4-5 of Algo. 2) and broadcasts the aggregated information back to all agents (line 6 of Algo. 2) to start the next iteration. We refer to those iterations between two communication rounds as an *epoch*.[2] So, our FN-UCB algorithm consists of a number of epochs which are separated by communication rounds.

Note that every agent $i$ only needs to train an NN in every communication round, i.e., only after the change in the log determinant of the covariance matrix of any agent exceeds a threshold $D$ (line 11 of Algo. 1). This has the additional benefit of reducing the computational cost due to the training of NNs. Interestingly, this is in a similar spirit as the adaptive batch size scheme in Gu et al. (2021) which only retrains the NN in Neural UCB after the change in the determinant of the covariance matrix exceeds a threshold and is shown to only slightly degrade the performance of Neural UCB.

### 3.2 THE TWO UPPER CONFIDENCE BOUNDS (UCBS)

Firstly, $\text{UCB}_{t,i}^a$ can be interpreted as the Linear UCB policy (Abbasi-Yadkori et al., 2011) using the neural tangent features $g(x;\theta_0)/\sqrt{m}$ as the input features. In iteration $t$, let $p$ denote the index of the

---

[2]The first (last) epoch is between a communication round and the beginning (end) of FN-UCB algorithm.

current epoch. Then, computing $\text{UCB}_{t,i}^a$ (line 5 of Algo. 1) makes use of two types of information. The first type of information, which *uses the observations from all $N$ agents before epoch $p$*, is used for computing $\text{UCB}_{t,i}^a$ via $W_{\text{sync}}$ and $B_{\text{sync}}$ (line 4 of Algo. 1). Specifically, as can be seen from line 5 of Algo. 2, $W_{\text{sync}}$ and $B_{\text{sync}}$ are computed by the central server by summing up the $W_{\text{new},i}$'s and $B_{\text{new},i}$'s from all agents (i.e., by aggregating the information from all agents) where $W_{\text{new},i}$ and $B_{\text{new},i}$ are computed using the local observations of agent $i$ (line 9 of Algo. 1). The second type of information used by $\text{UCB}_{t,i}^a$ (via $W_{\text{new},i}$ and $B_{\text{new},i}$ utilized in line 4 of Algo. 1) exploits the newly collected local observations of agent $i$ *in epoch $p$*. As a result, $\text{UCB}_{t,i}^a$ allows us to utilize the observations from all agents via the federated setting for accelerated exploration without requiring the agents to share their raw observations. $\text{UCB}_{t,i}^a$ is computed with the defined parameter $\nu_{TKN} \triangleq B + R[2(\log(3/\delta) + 1) + \widetilde{d}\log(1 + TKN/\lambda)]^{1/2}$ where $\delta \in (0,1)$.

Secondly, $\text{UCB}_{t,i}^b$ leverages the federated setting to improve the quality of NN for reward prediction (to achieve better exploitation) in a similar way to FedAvg, i.e., by averaging the parameters of the NNs trained by all agents using their local observations (McMahan et al., 2017). Specifically, when a communication round is started, every agent $i \in [N]$ uses its local observations $\mathcal{D}_{t,i} \triangleq \{(x_{\tau,i}, y_{\tau,i})\}_{\tau \in [t]}$ to train an NN (line 14 of Algo. 1). It uses initial parameters $\theta_0$ (i.e., shared among all agents (Sec. 2)) and trains the NN using gradient descent with learning rate $\eta$ for $J$ training iterations (see Theorem 1 for the choices of $\eta$ and $J$) to minimize the following loss function:

$$\mathcal{L}_{t,i}(\theta) \triangleq 0.5 \sum_{\tau=1}^t (f(x_{\tau,i}; \theta) - y_{\tau,i})^2 + 0.5m\lambda \|\theta - \theta_0\|_2^2 . \qquad (1)$$

The resulting NN parameters $\theta_t^i$'s from all $N$ agents are sent to the central server (line 16 of Algo. 1) who averages them (line 4 of Algo. 2) and broadcasts the aggregated $\theta_{\text{sync,NN}} \triangleq N^{-1} \sum_{i=1}^N \theta_t^i$ back to all agents to be used in the next epoch. In addition, to compute the second term of $\text{UCB}_{t,i}^b$, every agent needs to compute the matrix $V_{t,i}^{\text{local}}$ using its local inputs (line 10 of Algo. 1) and send its inverse to the central server (line 16 of Algo. 1) during a communication round; after that, the central server broadcasts these matrices $\{(V_i^{\text{local}})^{-1}\}_{i \in [N]}$ received from each agent back to all agents to be used in the second term of $\text{UCB}_{t,i}^b$ (line 6 of Algo. 1). Refer to Sec. 4.2 for a detailed explanation on the validity of $\text{UCB}_{t,i}^b$ as a high-probability upper bound on $h$ (up to additive error terms). $\text{UCB}_{t,i}^b$ is computed with the defined parameter $\nu_{TK} \triangleq B + R[2(\log(3N/\delta) + 1) + \widetilde{d}_{\max}\log(1 + TK/\lambda)]^{1/2}$.

## 3.3 Weight between the Two UCBs

Our choice of the weight $\alpha$ between the two UCBs, which naturally arises during our theoretical analysis (Sec. 4), has an interesting interpretation in terms of the relative strengths of the two UCBs and the exploration-exploitation trade-off. Specifically, $\widetilde{\sigma}_{t,i}^{\text{local}}(x) \triangleq \sqrt{\lambda} \left\| g(x; \theta_0)/\sqrt{m} \right\|_{(V_{t,i}^{\text{local}})^{-1}}$ intuitively represents our *uncertainty* about the reward at $x$ after conditioning on the local observations of agent $i$ up to iteration $t$ (Kassraie & Krause, 2022).[3] Next, $\widetilde{\sigma}_{t,i,\min}^{\text{local}} \triangleq \min_{x \in \mathcal{X}} \widetilde{\sigma}_{t,i}^{\text{local}}(x)$ and $\widetilde{\sigma}_{t,i,\max}^{\text{local}} \triangleq \max_{x \in \mathcal{X}} \widetilde{\sigma}_{t,i}^{\text{local}}(x)$ represent our smallest and largest uncertainties across the entire domain, respectively. Then, we choose $\alpha \triangleq \min_{i \in [N]} \alpha_{t,i}$ (line 4 of Algo. 2) where $\alpha_{t,i} \triangleq \widetilde{\sigma}_{t,i,\min}^{\text{local}}/\widetilde{\sigma}_{t,i,\max}^{\text{local}}$ (line 15 of Algo. 1). In other words, $\alpha_{t,i}$ is *the ratio between the smallest and largest uncertainty across the entire domain* for agent $i$, and $\alpha$ is the smallest such ratio $\alpha_{t,i}$ among all agents. Therefore, $\alpha$ is expected to be generally *increasing with the number of iterations/epochs*: $\widetilde{\sigma}_{t,i,\min}^{\text{local}}$ is already small after the first few iterations since the uncertainty at the queried contexts is very small; on the other hand, $\widetilde{\sigma}_{t,i,\max}^{\text{local}}$ is expected to be very large in early iterations and become smaller in later iterations only after a large number of contexts has been queried to sufficiently reduce the overall uncertainty in the entire domain. This implies that we give *more weight to $\text{UCB}_{t,i}^a$ in earlier iterations* and assign *more weight to $\text{UCB}_{t,i}^b$ in later iterations*. This, interestingly, turns out to have an intriguing practical interpretation: Relying more on $\text{UCB}_{t,i}^a$ in earlier iterations is reasonable because $\text{UCB}_{t,i}^a$ is able to utilize the observations from all agents to accelerate *exploration* in the early stage (Sec. 3.2); it is also sensible to give more emphasis to $\text{UCB}_{t,i}^b$ only in later iterations because the NN trained by every agent is only able to accurately model the reward function (for reliable *exploitation*) after it has collected enough observations to train its NN. In our practical implementation (Sec. 5), we will use the analysis here as an inspiration to design an increasing sequence of $\alpha$.

---

[3]Formally, $\widetilde{\sigma}_{t,i}^{\text{local}}(x)$ is the Gaussian process posterior standard deviation at $x$ conditioned on the local observations of agent $i$ till iteration $t$ and computed using the kernel $\widetilde{k}(x, x') = g(x; \theta_0)^\top g(x'; \theta_0)/m$.

### 3.4 COMMUNICATION COST

To achieve a better communication efficiency, we propose here a variant of our main `FN-UCB` algorithm called `FN-UCB (Less Comm.)` which differs from `FN-UCB` (Algos. 1 and 2) in two aspects. **Firstly**, the central server averages the matrices $\{(V_{t,i}^{\mathrm{local}})^{-1}\}_{i\in[N]}$ received from all agents to produce a single matrix $V_{\mathrm{sync,NN}}^{-1} = N^{-1}\sum_{i=1}^{N}(V_{t,i}^{\mathrm{local}})^{-1}$ and hence only broadcasts the single matrix $V_{\mathrm{sync,NN}}^{-1}$ instead of all $N$ received matrices $\{(V_{t,i}^{\mathrm{local}})^{-1}\}_{i\in[N]}$ to all agents (see line 6 of Algo. 2). **Secondly**, the $\mathrm{UCB}_{t,i}^{b}$ of every agent $i$ (line 6 of Algo. 1) is modified to use the matrix $V_{\mathrm{sync,NN}}^{-1}$: $\mathrm{UCB}_{t,i}^{b}(x) \triangleq f(x;\theta_{\mathrm{sync,NN}}) + \nu_{TK}\sqrt{\lambda}\big\|g(x;\theta_0)/\sqrt{m}\big\|_{V_{\mathrm{sync,NN}}^{-1}}$. To further reduce the communication cost of both `FN-UCB` and `FN-UCB (Less Comm.)` especially when the NN is large (i.e., its total number $p_0$ of parameters is large), we can follow the practice of previous works (Zhang et al., 2021; Zhou et al., 2020) to diagonalize the $p_0 \times p_0$ matrices, i.e., by only keeping the diagonal elements of the matrices. Specifically, we can diagonalize $W_{\mathrm{new},i}$ (line 9 of Algo. 1) and $V_{t,i}^{\mathrm{local}}$ (line 10 of Algo. 1), and let the central server aggregate only the diagonal elements of the corresponding matrices to obtain $W_{\mathrm{sync}}$ and $V_{\mathrm{sync,NN}}^{-1}$. This reduces both the communication and computational costs.

As a result, during a communication round, the parameters that an agent sends to the central server include $\{W_{\mathrm{new},i}, B_{\mathrm{new},i}, \theta_t^i, \alpha_{t,i}, (V_{t,i}^{\mathrm{local}})^{-1}\}$ (line 16 of Algo. 1) which constitute $p_0 + p_0 + p_0 + 1 + p_0 = \mathcal{O}(p_0)$ parameters and are the same for `FN-UCB` and `FN-UCB (Less Comm.)`. The parameters that the central server broadcasts to the agents include $\{W_{\mathrm{sync}}, B_{\mathrm{sync}}, \theta_{\mathrm{sync,NN}}, \alpha, \{(V_{t,i}^{\mathrm{local}})^{-1}\}_{i\in[N]}\}$ for `FN-UCB` (line 6 of Algo. 2) which amount to $p_0 + p_0 + p_0 + 1 + Np_0 = \mathcal{O}(Np_0)$ parameters. Meanwhile, `FN-UCB (Less Comm.)` only needs to broadcast $\mathcal{O}(p_0)$ parameters because the $N$ matrices $\{(V_{t,i}^{\mathrm{local}})^{-1}\}_{i\in[N]}$ are now replaced by a single matrix $V_{\mathrm{sync,NN}}^{-1}$. Therefore, the total number of exchanged parameters by `FN-UCB (Less Comm.)` is $\mathcal{O}(p_0)$ which is *comparable to the number of exchanged parameters in standard FL for supervised learning* (e.g., FedAvg) where the parameters (or gradients) of the NN are exchanged (McMahan et al., 2017). We will also analyze the total number of required communication rounds by `FN-UCB`, as well as by `FN-UCB (Less Comm.)`, in Sec. 4.1.

As we will discuss in Sec. 4.1, the variant `FN-UCB (Less Comm.)` has a looser regret upper bound than our main `FN-UCB` algorithm (Algos. 1 and 2). However, in practice, `FN-UCB (Less Comm.)` is recommended over `FN-UCB` because it achieves a very similar empirical performance as `FN-UCB` (which we have verified in Sec. 5.1) and yet incurs less communication cost.

## 4 THEORETICAL ANALYSIS

### 4.1 THEORETICAL RESULTS

**Regret Upper Bound.** For simplicity, we analyze the regret of a simpler version of our algorithm where we only choose the weight $\alpha$ using the method described in Sec. 3.3 in the first iteration after every communication round (i.e., first iteration of every epoch) and set $\alpha = 0$ in all other iterations. Note that when communication occurs after each iteration (i.e., when $D$ is sufficiently small), this version coincides with our original `FN-UCB` described in Algos. 1 and 2 (Sec. 3). The regret upper bound of `FN-UCB` is given by the following result (proof in Appendix C):

**Theorem 1.** *Let $\delta \in (0,1)$, $\lambda = 1 + 2/T$, and $D = \mathcal{O}(T/(N\widetilde{d}))$. Suppose that the NN width $m$ grows polynomially: $m \geq \mathrm{poly}(\lambda, T, K, N, L, \log(1/\delta), 1/\lambda_0)$. For the gradient descent training (line 14 of Algo. 1), let $\eta = C_4(m\lambda + mTL)^{-1}$ for some constant $C_4 > 0$ and $J = \widetilde{O}\big(TL/(\lambda C_4)\big)$. Then, with probability of at least $1 - \delta$, $R_T = \widetilde{O}\big(\widetilde{d}\sqrt{TN} + \widetilde{d}_{\max}N\sqrt{T}\big)$.*

Refer to Appendix C.1 for the detailed conditions on the NN width $m$ as well as the learning rate $\eta$ and number $J$ of iterations for the gradient descent training (line 14 of Algo. 1). Intuitively, the *effective dimension* $\widetilde{d}$ measures the actual underlying dimension of the set of all $TKN$ contexts for all agents (Zhang et al., 2021), and $\widetilde{d}_{\max} \triangleq \max_{i\in[N]}\widetilde{d}_i$ is the maximum among the underlying dimensions of the set of $TK$ contexts for each of the $N$ agents. Zhang et al. (2021) showed that if all contexts lie in a $d'$-dimensional subspace of the RKHS induced by the NTK, then the effective dimension of these contexts can be upper-bounded by the constant $d'$.

**The first term** $\widetilde{d}\sqrt{TN}$ in the regret upper bound (Theorem 1) arises due to $\mathrm{UCB}_{t,i}^{a}$ and reflects the benefit of the federated setting. In particular, this term matches the regret upper bound of standard Neural UCB (Zhou et al., 2020) running for $TN$ iterations and so, the average regret $\widetilde{d}\sqrt{T/N}$

across all agents decreases with a larger number $N$ of agents. **The second term** $\widetilde{d}_{\max} N \sqrt{T}$ results from $\text{UCB}_{t,i}^b$ which involves two major components of our algorithm: the use of NNs for reward prediction and the aggregation of the NN parameters. Although not reflecting the benefit of a larger $N$ in the regret bound, both components are important to our algorithm. Firstly, the use of NNs for reward prediction is a crucial component in neural contextual bandits in order to exploit the strong representation power of NNs. This is similar in spirit to the works on neural contextual bandits (Zhang et al., 2021; Zhou et al., 2020) in which the use of NNs for reward prediction does not improve the regret upper bound (compared with using the linear prediction given by the first term of $\text{UCB}_{t,i}^a$) and yet significantly improves the practical performance. Secondly, the aggregation of the NN parameters is also important for the performance of our FN-UCB since it allows us to exploit the federated setting in a similar way to FL for supervised learning which has been repeatedly shown to improve the performance (Kairouz et al., 2019). We have also empirically verified (Sec. 5.1) that both components (i.e., the use of NNs for reward prediction and the aggregation of NN parameters) are important to the practical performance of our algorithm. The work of Huang et al. (2021a) has leveraged the NTK to analyze the convergence of FedAvg for supervised learning (McMahan et al., 2017) which also averages the NN parameters in a similar way to our algorithm. Note that their convergence results also do not improve with a larger number $N$ of agents but in fact become worse with a larger $N$.

Of note, in the single-agent setting where $N = 1$, we have that $\widetilde{d} = \widetilde{d}_{\max}$ (Sec. 2). Therefore, our regret upper bound from Theorem 1 reduces to $R_T = \widetilde{O}(\widetilde{d}\sqrt{T})$, which, interestingly, matches the regret upper bounds of standard neural bandit algorithms including Neural UCB (Zhou et al., 2020) and Neural TS (Zhang et al., 2021). We also prove (App. C.7) that FN-UCB (Less Comm.), which is a variant of our FN-UCB with a better communication efficiency (Sec. 3.4), enjoys a regret upper bound of $R_T = \widetilde{O}(\widetilde{d}\sqrt{TN} + \widetilde{d}_{\max} N\sqrt{TN})$, whose second term is worse than that of FN-UCB (Theorem 1) by a factor of $\sqrt{N}$. In addition, we have also analyzed our general algorithm which *does not set* $\alpha = 0$ *in any iteration* (results and analysis in Appendix F), which requires an additional assumption and only introduces an additional multiplicative constant to the regret bound.

**Communication Complexity.** The following result (proof in App. D) gives a theoretical guarantee on the communication complexity of FN-UCB, including its variant FN-UCB (Less Comm.):

**Theorem 2.** *With the same parameters as Theorem 1, if the NN width $m$ satisfies $m \geq \text{poly}(T, K, N, L, \log(1/\delta))$, then with probability of at least $1 - \delta$, the total number of communication rounds for* FN-UCB *satisfies* $C_T = \widetilde{O}(\widetilde{d}\sqrt{N})$.

The specific condition on $m$ required by Theorem 2 corresponds to condition 1 listed in App. C.1 (see App. D for details) which is a subset of the conditions required by Theorem 1. Following the same discussion on the effective dimension $\widetilde{d}$ presented above, if all contexts lie in a $d'$-dimensional subspace of the RKHS induced by the NTK, then $\widetilde{d}$ can be upper-bounded by the constant $d'$, consequently leading to a communication complexity of $C_T = \widetilde{O}(\sqrt{N})$.

## 4.2 Proof Sketch

We give a brief sketch of our regret analysis for Theorem 1 (detailed proof in Appendix C). To begin with, we need to prove that both $\text{UCB}_{t,i}^a$ and $\text{UCB}_{t,i}^b$ are valid high-probability upper bounds on the reward function $h$ (App. C.3) given that the conditions on $m$, $\eta$, and $J$ in App. C.1 are satisfied.

Since $\text{UCB}_{t,i}^a$ can be viewed as Linear UCB using the neural tangent features $g(x; \theta_0)/\sqrt{m}$ as the input features (Sec. 3), its validity as a high-probability upper bound on $h$ can be proven following similar steps as that of standard linear and kernelized bandits (Chowdhury & Gopalan, 2017) (see Lemma 3 in App. C.3). Next, to prove that $\text{UCB}_{t,i}^b$ is also a high-probability upper bound on $h$ (up to additive error terms), let $\theta_{t,i}^{\text{local}} \triangleq (V_{t,i}^{\text{local}})^{-1}(\sum_{\tau=1}^{t} y_{\tau,i} g(x_{\tau,i}; \theta_0)/\sqrt{m})$ which is defined in the same way as $\overline{\theta}_{t,i}$ (line 4 of Algo. 1) except that $\theta_{t,i}^{\text{local}}$ only uses the local observations of agent $i$. **Firstly**, we show that $f(x; \theta_{\text{sync,NN}})$ (i.e., the NN prediction using the aggregated parameters) is close to $N^{-1}\sum_{i=1}^{N}\langle g(x; \theta_0)/\sqrt{m}, \theta_{t,i}^{\text{local}}\rangle$ which is the linear prediction using $\theta_{t,i}^{\text{local}}$ averaged over all agents. This is achieved by showing that the linear approximation of the NN at $\theta_0$ is close to both terms. **Secondly**, we show that the absolute difference between the linear prediction $\langle g(x; \theta_0)/\sqrt{m}, \theta_{t,i}^{\text{local}}\rangle$ of agent $i$ and the reward function $h(x)$ can be upper-bounded by $\nu_{TK}\sqrt{\lambda}||g(x; \theta_0)/\sqrt{m}||_{(V_{t,i}^{\text{local}})^{-1}}$. This can be done following similar steps as the proof for $\text{UCB}_{t,i}^a$ mentioned above. **Thirdly**, using the

averaged linear prediction $N^{-1} \sum_{i=1}^{N} \langle g(x; \theta_0)/\sqrt{m}, \theta_{t,i}^{\text{local}} \rangle$ as an intermediate term, the difference between $f(x; \theta_{\text{sync,NN}})$ and $h(x)$ can be upper-bounded. This implies the validity of $\text{UCB}_{t,i}^b$ as a high-probability upper bound on $h$ (up to additive error terms which are small given the conditions on $m$, $\eta$, and $J$ presented in App. C.1), as formalized by Lemma 4 in App. C.3.

Next, following similar footsteps as the analysis in Wang et al. (2020), we separate all epochs into "good" epochs (intuitively, those epochs during which the amount of newly collected information from all agents is not too large) and "bad" epochs (details in App. C.2), and then separately upper-bound the regrets incurred in these two types of epochs. For good epochs (App. C.4), we are able to derive a tight upper bound on the regret $r_{t,i} = h(x_{t,i}^*) - h(x_{t,i})$ in each iteration $t$ by making use of the fact that the change of information in a good epoch is bounded, and consequently obtain a tight upper bound on the total regrets in all good epochs. For bad epochs (App. C.5), we make use of the result from App. C.2 which guarantees that the total number of bad epochs can be upper-bounded. As a result, with an appropriate choice of $D = \mathcal{O}(T/(N\widetilde{d}))$, the growth rate of the total regret incurred in bad epochs is smaller than that in good epochs. Lastly, the final regret upper bound follows from adding up the total regrets from good and bad epochs (App. C.6).

## 5 EXPERIMENTS

All figures in this section plot the average cumulative regret across all $N$ agents up to an iteration, which allows us to inspect the benefit that the federated setting brings to an agent (on average). In all presented results, unless specified otherwise (by specifying a value of $D$), a communication round happens after each iteration. All curves stand for the mean and standard error from 3 independent runs. Some experimental details and results are deferred to App. E due to space limitation.

### 5.1 SYNTHETIC EXPERIMENTS

We firstly use synthetic experiments to illustrate some interesting insights about our algorithm. Similar to that of Zhou et al. (2020), we adopt the synthetic functions of $h(x) = \cos(3\langle a, x \rangle)$ and $h(x) = 10(\langle a, x \rangle)^2$ which are referred to as the `cosine` and `square` functions, respectively. We add a Gaussian observation noise with a standard deviation of $0.01$. The parameter $a$ is a 10-dimensional vector randomly sampled from the unit hypersphere. In each iteration, every agent receives $K = 4$ contexts (arms) which are randomly sampled from the unit hypersphere. For fair comparisons, for all methods (including our `FN-UCB`, Neural UCB, and Neural TS), we use the same set of parameters of $\lambda = \nu_{TKN} = \nu_{TK} = 0.1$ and use an NN with 1 hidden layer and a width of $m = 20$. As suggested by our theoretical analysis (Sec. 3.3), we select an increasing sequence of $\alpha$ which is linearly increasing (to 1) in the first 700 iterations, and let $\alpha = 1$ afterwards. To begin with, we compare our main `FN-UCB` algorithm and its variant `FN-UCB (Less Comm.)` (Sec. 3.4). The results (Figs. 3a and 3b in App. E) show that their empirical performances are very similar. So, for practical deployment, we recommend the use of `FN-UCB (Less Comm.)` as it is more communication-efficient and achieves a similar performance. Accordingly, we will use the variant `FN-UCB (Less Comm.)` in all our subsequent experiments and refer to it as `FN-UCB` for simplicity.

Fig. 1 presents the results. Figs. 1a and 1b show that our `FN-UCB` with $N = 1$ agent performs comparably with Neural UCB and Neural TS, and that the federation of a larger number $N$ of agents consistently improves the performance of our `FN-UCB`. Note that the federation of $N = 2$ agents can already provide significant improvements over non-federated algorithms. Fig. 1c gives an illustration of the importance of different components in our `FN-UCB`. The red curve is obtained by removing $\text{UCB}_{t,i}^b$ (i.e., letting $\alpha = 0$) and the green curve corresponds to removing $\text{UCB}_{t,i}^a$. The red curve shows that relying solely on $\text{UCB}_{t,i}^a$ leads to significantly larger regrets in the long run due to its inability to utilize NNs to model the reward functions. On the other hand, the green curve incurs larger regrets than the red curve initially; however, after more observations are collected (i.e., after the NNs are trained with enough data to accurately model the reward function), it quickly learns to achieve much smaller regrets. These results provide empirical justifications for our discussion on the weight between the two UCBs (Sec. 3.3): It is reasonable to use an increasing sequence of $\alpha$ such that more weight is given to $\text{UCB}_{t,i}^a$ initially and then to $\text{UCB}_{t,i}^b$ later. The yellow curve is obtained by removing the step of aggregating (i.e., averaging) the NN parameters (in line 4 of Algo. 2), i.e., when calculating $\text{UCB}_{t,i}^b$ (line 6 of Algo. 1), we use $\theta_t^i$ to replace $\theta_{\text{sync,NN}}$. The results show that the aggregation of the NN parameters significantly improves the performance of `FN-UCB` (i.e., the blue curve has much smaller regrets than the yellow one) and is hence an indispensable part of our

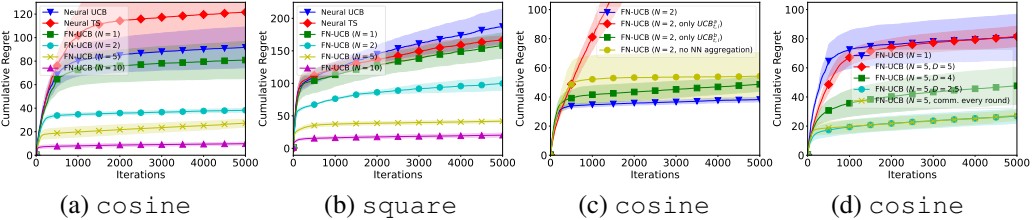

(a) cosine     (b) square     (c) cosine     (d) cosine

Figure 1: Cumulative regret with varying number of agents for the (a) `cosine` function and (b) `square` function. (c) Illustration of the importance of different components of our `FN-UCB` algorithm (`cosine` function). (d) Performances with different values of $D$ (`cosine` function). The average number of rounds of communications are $348.0, 380.0, 456.7$ for $D = 5, 4, 2.5$, respectively.

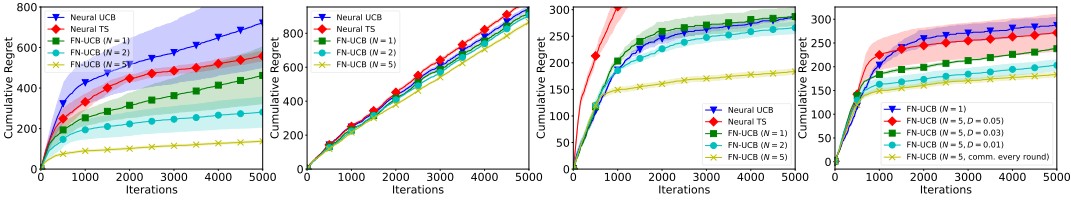

(a) shuttle    (b) magic telescope    (c) shuttle (diag.)    (d) shuttle (diag.)

Figure 2: Results ($m = 20$) for (a) `shuttle` and (b) `magic`. (c) Results for `shuttle` with diagonal approximation ($m = 50$). (d) Results for `shuttle` with different values of $D$. The average number of communication rounds are $3850.7, 4442.7, 4906.3$ for $D = 0.05, 0.03, 0.01$, respectively.

`FN-UCB`. Lastly, Fig. 1d shows that more frequent communications (i.e., smaller values of $D$ which make it easier to initiate a communication round; see line 11 of Algo. 1) lead to smaller regrets.

## 5.2 REAL-WORLD EXPERIMENTS

We adopt the `shuttle` and `magic telescope` datasets from the UCI machine learning repository (Dua & Graff, 2017) and construct the experiments following a widely used protocol in previous works (Li et al., 2010a; Zhang et al., 2021; Zhou et al., 2020). A $K$-class classification problem can be converted into a $K$-armed contextual bandit problem. In each iteration, an input $\mathbf{x}$ is randomly drawn from the dataset and is then used to construct $K$ context feature vectors $\mathbf{x}_1 = [\mathbf{x}; \mathbf{0}_d; \dots; \mathbf{0}_d], \mathbf{x}_2 = [\mathbf{0}_d; \mathbf{x}; \dots; \mathbf{0}_d], \dots, \mathbf{x}_K = [\mathbf{0}_d; \dots; \mathbf{0}_d; \mathbf{x}]$ which correspond to the $K$ classes. The reward is 1 if the arm with the correct class is pulled, and 0 otherwise. For fair comparisons, we use the same set of parameters of $\lambda = 10$, $\nu_{TKN} = 0.1$, and $\nu_{TK} = 0.01$ for all methods. Figs. 2a and 2b present the results for the two datasets (1 hidden layer, $m = 20$) and show that our `FN-UCB` with $N = 2$ agents consistently outperforms standard Neural UCB and Neural TS, and its performance also improves with the federation of more agents. Fig. 2c shows the results for `shuttle` when diagonal approximation (Sec. 3.4) is applied to the NNs (1 hidden layer, $m = 50$); the corresponding results are consistent with those in Fig. 2a.[4] Moreover, the regrets in Fig. 2c are in general smaller than those in Fig. 2a. This may suggest that in practice, a wider NN with diagonal approximation may be preferable to a narrower NN without diagonal approximation since it not only improves the performance but also reduces the computational and communication costs (Sec. 3.4). Fig. 2d plots the regrets of `shuttle` (with diagonal approximation) for different values of $D$ and shows that more frequent communications lead to better performances and are hence consistent with that in Fig. 1d. For completeness, we also compare their performance with that of linear and kernelized contextual bandit algorithms (for the experiments in both Secs. 5.1 and 5.2), and the results (Fig. 4, App. E) show that they are outperformed by neural contextual bandit algorithms.

## 6 CONCLUSION

This paper describes the first federated neural contextual bandit algorithm called `FN-UCB`. We use a weighted combination of two UCBs and the choice of this weight required by our theoretical analysis has an interesting interpretation emphasizing accelerated exploration initially and accurate prediction of the aggregated NN later. We derive upper bounds on the regret and communication complexity of `FN-UCB`, and verify its effectiveness using empirical experiments. Our algorithm is not equipped with privacy guarantees, which may be a potential limitation and will be tackled in future work.

---

[4]Since diagonalization increases the scale of the first term in $\text{UCB}_{t,i}^a$, we use a heuristic to rescale the values of this term for all contexts such that the max and min values (among all contexts) are 0 and 1 after rescaling.

## REPRODUCIBILITY STATEMENT

We have included the necessary details to ensure the reproducibility of our theoretical and empirical results. For our theoretical results, we have stated all our assumptions in Sec. 2, added a proof sketch in Sec. 4.2, and included the complete proofs in App. C and App. D. Our detailed experimental settings have been described in Sec. 5.1, Sec. 5.2, and App. E. Our code has been submitted as supplementary material.

### ACKNOWLEDGMENTS

This research/project is supported by A*STAR under its RIE2020 Advanced Manufacturing and Engineering (AME) Industry Alignment Fund – Pre Positioning (IAF-PP) (Award A19E4a0101).

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

## A    RELATED WORKS

**Federated Bandits.**    Federated learning (FL) has received enormous attention in recent years (Kairouz et al., 2019; Li et al., 2021; 2014; McMahan et al., 2017). A number of recent works have extended the classic $K$-armed bandits (i.e., the arms are not associated with feature vectors) to the federated setting. Li & Song (2022) and Li et al. (2020) focused on incorporating privacy guarantees into federated $K$-armed bandits in both centralized and decentralized settings. Shi & Shen (2021) proposed a setting where the goal is to minimize the regret of a global bandit whose reward of an arm is the average of the rewards of the corresponding arm from all agents, which was later extended by adding personalization such that every agent aims to maximize a weighted combination between the global and local rewards (Shi et al., 2021). Subsequent works on federated $K$-armed bandits have focused on other important aspects such as decentralized communication via the gossip algorithm (Zhu et al., 2021b), the security aspect via cryptographic techniques (Ciucanu et al., 2022), uncoordinated exploration (Yan et al., 2022), and robustness against Byzantine attacks (Demirel et al., 2022). Regarding federated linear contextual bandits, Wang et al. (2020) proposed a distributed linear contextual bandit algorithm which allows every agent to use the observations from the other agents by only exchanging the sufficient statistics to calculate the Linear UCB policy. Subsequently, Dubey & Pentland (2020) extended the method from Wang et al. (2020) to consider differential privacy and decentralized communication, Huang et al. (2021b) considered a setting where every agent is associated with a unique context vector, Li & Wang (2022a) focused on asynchronous communication, and Jadbabaie et al. (2022) considered the robustness against Byzantine attacks. Federated kernelized/GP bandits (also named federated Bayesian optimization) have been explored by Dai et al. (2020; 2021), which focused on the practical problem of hyperparameter tuning in the federated setting. The recent works of Li et al. (2022); Li & Wang (2022b) have, respectively, focused on deriving communication-efficient algorithms for federated kernelized and generalized linear bandits. In addition to federated bandits, other similar sequential decision-making problems have also been extended to the federated setting, such as federated reinforcement learning (Fan et al., 2021; Zhuo et al., 2019) and federated hyperparameter tuning (Holly et al., 2021; Khodak et al., 2021; Zhou et al., 2021).

**Neural Bandits.**    Since the pioneering works of Zhou et al. (2020) and Zhang et al. (2021) which, respectively, introduced Neural UCB and Neural TS, a number of recent works have focused on different aspects of neural contextual bandits. Xu et al. (2020) reduced the computational cost of Neural UCB by using an NN as a feature extractor and applying Linear UCB only to the last layer of the learned NN, Kassraie & Krause (2022) analyzed the maximum information gain of the NTK and hence derived no-regret algorithms, Gu et al. (2021) focused on the batch setting in which the policy is only updated at a small number of time steps, Nabati et al. (2021) aimed to reduce the memory requirement of Neural UCB, Lisicki et al. (2021) performed an empirical investigation of neural bandit algorithms to verify their practical effectiveness, Ban et al. (2022) adopted a separate NN for exploration in neural contextual bandits, Ban & He (2021) applied the convolutional NTK, Jia et al. (2021) used perturbed rewards to train the NN to remove the need for explicit exploration, Nguyen-Tang et al. (2022) incorporated offline policy learning into neural contextual bandits, Zhu et al. (2021a) studied pure exploration in kernel and neural bandits, Kassraie et al. (2022) applied graph NNs in neural bandits to handle graph-structured data, Salgia et al. (2022) extended neural bandits beyond the ReLU activation to consider smoother activation functions, and Dai et al. (2022) introduced a scalable batch Neural TS algorithm through sample-then-optimize optimization.

## B    MORE BACKGROUND

In this section, we give more details on some of the technical background mentioned in Sec. 2. The details in this section all follow the works of Zhang et al. (2021); Zhou et al. (2020), and we present them here for completeness.

**Definition of the NN** $f(x;\theta)$**.**   Let $\mathbf{W}_1 \in \mathbb{R}^{m \times d}$, $\mathbf{W}_l \in \mathbb{R}^{m \times m}, \forall l = 2, \ldots, L-1$, and $\mathbf{W}_L \in \mathbb{R}^{m \times 1}$, then the NN $f(x;\theta)$ is defined as

$$f_1 = \mathbf{W}_1 x,$$
$$f_l = \mathbf{W}_l \text{ReLU}(f_{l-1}), \forall l = 2 \ldots, L,$$
$$f(x;\theta) = \sqrt{m} f_L,$$

in which $\text{ReLU}(z) = \max(z, 0)$ denotes the rectified linear unit (ReLU) activation function and is applied to each element of $f_{l-1}$. With this definition of the NN, $\theta$ denotes the collection of all parameters of the NN: $\theta = (\text{vec}(\mathbf{W}_1), \ldots, \text{vec}(\mathbf{W}_L)) \in \mathbb{R}^{p_0}$.

**Details of the Initialization Scheme** $\theta_0 \sim \text{init}(\cdot)$**.**   To obtain the initial parameters $\theta_0$, for each $l = 1, \ldots, L-1$, let $\mathbf{W}_l = \begin{pmatrix} \mathbf{W} & \mathbf{0} \\ \mathbf{0} & \mathbf{W} \end{pmatrix}$ where each entry of $\mathbf{W}$ is independently sampled from $\mathcal{N}(0, 4/m)$, and let $\mathbf{W}_L = (\mathbf{w}^\top, -\mathbf{w}^\top)$ where each entry of $\mathbf{w}$ is independently sampled from $\mathcal{N}(0, 2/m)$. This initialization scheme is the same as that used by the works of Zhang et al. (2021); Zhou et al. (2020).

**Definitions of the NTK Matrices H and** $\mathbf{H}_i$**'s.**   To simplify the exposition here, we use $\{x^j\}_{j=1,\ldots,TKN}$ to denote the set of all contexts from all iterations, all arms and *all agents*: $\{x_{t,i}^k\}_{t \in [T], k \in [K], i \in [N]}$. We can then define

$$\widetilde{\mathbf{H}}_{i,j}^{(1)} = \boldsymbol{\Sigma}_{i,j}^{(1)} = \langle x^i, x^j \rangle, \mathbf{A}_{i,j}^{(l)} = \begin{pmatrix} \boldsymbol{\Sigma}_{i,i}^{(l)} & \boldsymbol{\Sigma}_{i,j}^{(l)} \\ \boldsymbol{\Sigma}_{i,j}^{(l)} & \boldsymbol{\Sigma}_{j,j}^{(l)} \end{pmatrix},$$

$$\boldsymbol{\Sigma}_{i,j}^{(l+1)} = 2 \mathbb{E}_{(u,v) \sim \mathcal{N}(\mathbf{0}, \mathbf{A}_{i,j}^{(l)})} \max(u, 0) \max(v, 0),$$

$$\widetilde{\mathbf{H}}_{i,j}^{(l+1)} = 2 \widetilde{\mathbf{H}}_{i,j}^{(l)} \mathbb{E}_{(u,v) \sim \mathcal{N}(\mathbf{0}, \mathbf{A}_{i,j}^{(l)})} \mathbb{1}(u > 0) \mathbb{1}(v > 0) + \boldsymbol{\Sigma}_{i,j}^{(l+1)}.$$

With these definitions, the NTK matrix is defined as $\mathbf{H} = (\widetilde{\mathbf{H}}^{(L)} + \boldsymbol{\Sigma}^{(L)})/2$. Similarly, $\mathbf{H}_i$ can be obtained in the same way by only using all contexts from agent $i$ in the definitions above, i.e., now we use $\{x^j\}_{j=1,\ldots,TK}$ to denote $\{x_{t,i}^k\}_{t \in [T], k \in [K]}$ and plug these $TK$ contexts into the definitions above to obtain $\mathbf{H}_i$.

## C    PROOF OF REGRET UPPER BOUND (THEOREM 1)

We use $p$ to index different epochs and denote by $P$ the total number of epochs. We use $t_p$ to denote the first iteration of epoch $p$, and use $E_p$ to represent the length (i.e., number of iterations) of epoch $p$. Throughout our theoretical analysis, we will denote different error probabilities as $\delta_1, \ldots, \delta_6$, which we will combine via a union bound at the end of the proof to ensure that our final regret upper bound holds with probability of at least $1 - \delta$.

### C.1    CONDITIONS ON THE WIDTH $m$ OF THE NEURAL NETWORKS

We list here the detailed conditions on the width $m$ of the NN that are needed by our theoretical analysis. These include two types of conditions, some of them (conditions 1-4) are required for our regret upper bound to hold (i.e., they are used during the proof to derive the regret upper bound), whereas the others (conditions 5-6) are used after the final regret upper bound is derived to ensure that the final regret upper bound is small (see App. C.6).

When presenting our detailed proofs starting from the next subsection, we will refer to each of these conditions whenever they are used by the corresponding lemmas. Different lemmas may use different leading constants in their required condition (i.e., lower bound) on $m$, but here we use the same constant $C > 0$ for all lower bounds for simplicity, which can be considered as simply taking the maximum among all these different leading constants for different lemmas.

  1. $m \geq CT^6 K^6 N^6 L^6 \log(TKNL/\delta)$,
  2. $m \geq CT^4 K^4 N^4 L^6 \log(T^2 K^2 N^2 L/\delta)/\lambda_0^4$,

3. $m \geq C\sqrt{\lambda}L^{-3/2}[\log(TKNL^2/\delta)]^{3/2}$,

4. $m(\log m)^{-3} \geq CTL^{12}\lambda^{-1} + CT^7\lambda^{-8}L^{18}(\lambda + LT)^6 + CL^{21}T^7\lambda^{-7}(1 + \sqrt{T/\lambda})^6$.

5. $m(\log m)^{-3} \geq CT^{10}N^6\lambda^{-4}L^{18}$,

6. $m(\log m)^{-3} \geq CT^{16}N^6L^{24}\lambda^{-10}(1 + \sqrt{T/\lambda})^6$.

Some of these conditions above can be combined, but we leave them as separate conditions to make it easier to refer to the corresponding place in the proof where a particular condition is needed.

Furthermore, to achieve a small upper bound on the cumulative regret, we also need to place some conditions on the learning rate $\eta$ and number of iterations $J$ for the gradient descent training (line 14 of Algo. 1). Specifically, we need to choose the learning rate as

$$\eta = C_4(m\lambda + mTL)^{-1}, \tag{2}$$

in which $C_4 > 0$ is an absolute constant such that $C_4 \leq 1 + TL$, and choose

$$J = \frac{1}{C_4}\left(1 + \frac{TL}{\lambda}\right)\log\left(\frac{1}{3C_2N}\sqrt{\frac{\lambda}{T^3L}}\right) = \widetilde{O}\left(TL/(\lambda C_4)\right). \tag{3}$$

### C.2 DEFINITION OF GOOD AND BAD EPOCHS

Denote the matrix $V_{\text{last}}$ (see line 18 of Algo. 1) after epoch $p$ as $V_p$. As a result, the matrix $V_P$ is calculated using all selected inputs from all agents: $V_P = \sum_{t=1}^{T}\sum_{i=1}^{N} g(x_{t,i};\theta_0)g(x_{t,i};\theta_0)^\top/m + \lambda I$. Define $V_0 \triangleq \lambda I$. Imagine that we have a hypothetical agent which chooses all $T \times N$ queries $\{x_{t,i}\}_{t\in[T],i\in[N]}$ sequentially in a round-robin fashion (i.e., the hypothetical agent chooses $x_{1,1}, x_{1,2}, \ldots, x_{2,1}, x_{2,2}, \ldots, x_{T,N}$), and denote the corresponding hypothetical covariance matrix as $\widetilde{V}_{t,i} = \sum_{\tau=1}^{t-1}\sum_{j=1}^{N} g(x_{\tau,j};\theta_0)g(x_{\tau,j};\theta_0)^\top/m + \sum_{j=1}^{i} g(x_{t,j};\theta_0)g(x_{t,j};\theta_0)^\top/m + \lambda I$. We represent the indices of this hypothetical agent by $t' \in [TN]$ to distinguish it from our original multi-agent setting. Define $J_{TN} \triangleq [g(x_{t'};\theta_0)]_{t'\in[TN]}$ which is a $p_0 \times (TN)$ matrix, and define $\mathbf{K}_{TN} \triangleq J_{TN}^\top J_{TN}/m$, which is a $(TN) \times (TN)$ matrix. According to thes definitions, we have that

**Lemma 1** (Lemma B.7 of Zhang et al. (2021)). *Let* $\delta_1 \in (0,1)$. *If* $m \geq CT^6N^6K^6L^6\log(TNKL/\delta_1)$, *we have with probability of at least* $1 - \delta_1$ *that*

$$\log\det(I + \lambda^{-1}\mathbf{K}_{t'}) \leq \log\det(I + \lambda^{-1}\mathbf{H}) + 1, \forall t' \in [TN].$$

The condition on $m$ given in Lemma 1 corresponds to condition 1 listed in App. C.1, except that $\delta_1$ is used here instead of $\delta$ in App. C.1. Lemma 1 allows us to derive the following equation, which we will use (at the end of this section) to justify that the total number of "bad" epochs is not too large.

$$
\begin{aligned}
\sum_{p=0}^{P-1}\log\frac{\det V_{p+1}}{\det V_p} = \log\frac{\det V_P}{\det V_0} &\overset{(a)}{=} \log\frac{\det\left(J_{TN}J_{TN}^\top/m + \lambda I\right)}{\det V_0} \\
&= \log\frac{\det\left(\lambda\left(\lambda^{-1}J_{TN}J_{TN}^\top/m + I\right)\right)}{\det V_0} \\
&\overset{(b)}{=} \log\frac{\lambda^{p_0}\det\left(\lambda^{-1}J_{TN}J_{TN}^\top/m + I\right)}{\lambda^{p_0}} \\
&= \log\det\left(\lambda^{-1}J_{TN}J_{TN}^\top/m + I\right) \\
&\overset{(c)}{=} \log\det\left(\lambda^{-1}J_{TN}^\top J_{TN}/m + I\right) \\
&= \log\det\left(\lambda^{-1}\mathbf{K}_{TN} + I\right) \\
&\overset{(d)}{\leq} \log\det\left(\lambda^{-1}\mathbf{H} + I\right) + 1 \\
&\overset{(e)}{=} \widetilde{d}\log(1 + TKN/\lambda) + 1 \triangleq R'.
\end{aligned} \tag{4}
$$

Step $(a)$ is because $V_P = J_{TN} J_{TN}^\top / m + \lambda I$ according to our definition of $J_{TN}$ above. Step $(b)$ follows from our definition of $V_0 = \lambda I$ above, as well as some standard properties of matrix determinant. Step $(c)$ follows because: $\det(\mathbf{A}\mathbf{A}^\top + I) = \det(\mathbf{A}^\top \mathbf{A} + I)$. Step $(d)$ has made use of Lemma 1 above, which suggests that equation 4 holds with probability of at least $1 - \delta_1$. Step $(e)$ follows from the definition of $\widetilde{d} \triangleq \frac{\log \det(I + \mathbf{H}/\lambda)}{\log(1 + TKN/\lambda)}$ (Sec. 2). In the last step, we have defined $R' \triangleq \widetilde{d} \log(1 + TKN/\lambda) + 1$. We further define $\overline{R} \triangleq \lceil R' \rceil$, in which $\lceil \cdot \rceil$ denotes the ceiling operator.

Now we define all epochs $p$'s which satisfy the following condition as "good epochs":

$$1 \leq \frac{\det V_p}{\det V_{p-1}} \leq e, \tag{5}$$

and define all other epochs as "bad epochs". The first inequality trivially holds for all epochs according to the way in which the matrices are constructed. It is easy to verify that the second inequality holds for at least $R$ epochs (with probability of at least $1 - \delta_1$). This is because if the second inequality is violated for more than $\overline{R}$ epochs (i.e., if $\log \frac{\det V_p}{\det V_{p-1}} > 1$ for more than $\overline{R}$ epochs), then $\sum_{p=0}^{P-1} \log \frac{\det V_{p+1}}{\det V_p} > \overline{R}$, which violates equation 4. This suggests that *there are no more than $\overline{R}$ bad epochs* (with probability of at least $1 - \delta_1$). From here onwards, we will denote the set of good epochs by $\mathcal{E}^{\text{good}}$ and the set of bad epochs by $\mathcal{E}^{\text{bad}}$.

## C.3   VALIDITY OF THE UPPER CONFIDENCE BOUND

In this section, we prove that the upper confidence bound used in our algorithm, $(1 - \alpha_t)\text{UCB}_{t,i}^a(x) + \alpha_t \text{UCB}_{t,i}^b(x)$ (used in line 7 of Algo. 1), is a valid high-probability upper bound on the reward function $h$. We will achieve this by separately proving that $\text{UCB}_{t,i}^a$ and $\text{UCB}_{t,i}^b$ are valid high-probability upper bounds on $h$ in the next two sections. Note that for both UCBs, unlike Neural UCB (Zhou et al., 2020) and Neural TS (Zhang et al., 2021) which use $\theta_t$ (the parameters of trained NNs) to calculate the exploration term (the second terms of $\text{UCB}_{t,i}^a$ and $\text{UCB}_{t,i}^b$), we instead use $\theta_0$. This is consistent with Kassraie & Krause (2022) who have shown that the use of $\theta_0$ gives accurate uncertainty estimation.

### C.3.1   VALIDITY OF $\text{UCB}_{t,i}^a$ AS A HIGH-PROBABILITY UPPER BOUND ON $h$:

To begin with, we will need the following lemma from Zhang et al. (2021).

**Lemma 2** (Lemma B.3 of Zhang et al. (2021)). *Let $\delta_2 \in (0, 1)$. There exists a constant $C > 0$ such that if*

$$m \geq CT^4 K^4 N^4 L^6 \log(T^2 K^2 N^2 L / \delta_2)/\lambda_0^4,$$

*then with probability of $\geq 1 - \delta_2$ over random initializations of $\theta_0$, there exists a $\theta^* \in \mathbb{R}^{p_0}$ such that*

$$h(x) = \langle g(x; \theta_0), \theta^* - \theta_0 \rangle, \quad \sqrt{m} \|\theta^* - \theta_0\|_2 \leq \sqrt{2\mathbf{h}^\top \mathbf{H}^{-1}\mathbf{h}} \leq B, \forall x \in \mathcal{X}_{t,i}, t \in [T], i \in [N]. \tag{6}$$

The condition on $m$ required by Lemma 2 corresponds to condition 2 listed in App. C.1, except that $\delta_2$ is used here instead of $\delta$ as in App. C.1. The following lemma formally guarantees the validity of $\text{UCB}_{t,i}^a$ as a high-probability upper-bound on $h$.

**Lemma 3.** *Let $\delta_3 \in (0, 1)$ and $\nu_{TKN} = B + R\sqrt{2(\log(1/\delta_3) + 1) + \widetilde{d}\log(1 + TKN/\lambda)}$. We have with probability of at least $1 - \delta_1 - \delta_2 - \delta_3$ for all $t \in [T], i \in [N]$, that*

$$|h(x) - \langle g(x; \theta_0)/\sqrt{m}, \overline{\theta}_{t,i} \rangle| \leq \nu_{TKN} \sqrt{\lambda} \big\| g(x; \theta_0)/\sqrt{m} \big\|_{\overline{V}_{t,i}^{-1}}, \forall x \in \mathcal{X}_{t,i}$$

*Proof.* Lemma 3 can be proved by following similar steps as the proof of Lemma 4.3 in the work of Zhang et al. (2021). Specifically, the proof of Lemma 3 requires Lemmas B.3, B.6 and B.7 of Zhang et al. (2021) (after being adapted for our setting). The adapted versions of Lemmas B.3 and B.7 of Zhang et al. (2021) have been presented in our Lemma 2 and Lemma 1, respectively. Of note, Lemma 1 and Lemma 2 require some conditions on the width $m$ of the NN, which have been listed as conditions 1 and 2 in Appendix C.1. Lastly, Lemma B.6 of Zhang et al. (2021), which makes use

of Theorem 1 of Chowdhury & Gopalan (2017), can be directly applied in our setting and introduces an error probability of $\delta_3$ (which appears in the expression of $\nu_{TKN}$). As a result, Lemma 3 holds with probability of at least $1 - \delta_1 - \delta_2 - \delta_3$, in which the error probabilities come from Lemma 1 ($\delta_1$), Lemma 2 ($\delta_2$) and the application of Lemma B.6 of Zhang et al. (2021) ($\delta_3$).

$\square$

### C.3.2 Validity of $\mathrm{UCB}_{t,i}^b$ as a High-Probability Upper Bound on $h$:

Note that $\mathrm{UCB}_{t,i}^b$ is updated only in every communication round. We denote the set of iterations after which $\mathrm{UCB}_{t,i}^b$ is updated (i.e., the last iteration in every epoch) as $\mathcal{T}_{-1} \triangleq \{t_p - 1\}_{p=2,\ldots,P-1}$, which immediately implies that $\mathcal{T}_{-1} \subset [T]$ and hence $|\mathcal{T}_{-1}| \leq T$.

**Lemma 4.** *Let $\delta_4, \delta_5 \in (0,1)$, and $\nu_{TK} = B + R\sqrt{2(\log(N/\delta_4) + 1) + \widetilde{d}_{\max}\log(1 + TK/\lambda)}$ Suppose the width $m$ of the NN satisfies $m \geq C\sqrt{\lambda}L^{-3/2}[\log(TKNL^2/\delta_5)]^{3/2}$ for some constant $C > 0$, as well as condition 4 in Appendix C.1. Suppose the learning rate $\eta$ and number of iterations $J$ of the gradient descent training satisfy the conditions in equation 2 and equation 3 (App. C.1), respectively. We have with probability of at least $1 - \delta_4 - \delta_5$ for all $t \in \mathcal{T}_{-1}, i \in [N]$, that*

$$|h(x) - f(x;\theta_{sync,NN})| \leq \nu_{TK}\sqrt{\lambda}\frac{1}{N}\sum_{i=1}^{N}\left\|g(x;\theta_0)/\sqrt{m}\right\|_{(V_i^{local})^{-1}} + \varepsilon_{linear}(m,T), \forall x \in \mathcal{X}_{t,i}.$$

*Proof.* Note that the condition on $m$ listed in the lemma, $m \geq C\sqrt{\lambda}L^{-3/2}[\log(TKNL^2/\delta_5)]^{3/2}$, corresponds to condition 3 listed in App. C.1 except that $\delta_5$ is used here instead of $\delta$. Therefore, the validity of Lemma 4 requires conditions 3 and 4 on $m$ (App. C.1) to be satisfied. For ease of exposition, we separate our proof into three steps.

**Step 1: NN Output $f(x;\theta_{\text{sync,NN}})$ Is Close to (Averaged) Linear Prediction**

Based on Lemma C.2 of Zhang et al. (2021), if the conditions on $m$ listed in Lemma 4 is satisfied, then for any $\widetilde{\theta}$ such that $\left\|\widetilde{\theta} - \theta_0\right\|_2 \leq 2\sqrt{t/(m\lambda)}$, there exists a constant $C_1 > 0$ such that we have with probability of at least $1 - \delta_5$ over random initializations $\theta_0$ that

$$\begin{aligned}
|f(x;\widetilde{\theta}) - \langle g(x;\theta_0), \widetilde{\theta} - \theta_0 \rangle| &\leq C_1 t^{2/3}m^{-1/6}\lambda^{-2/3}L^3\sqrt{\log m} \\
&\leq C_1 T^{2/3}m^{-1/6}\lambda^{-2/3}L^3\sqrt{\log m} \qquad (7) \\
&\triangleq \varepsilon_{\text{linear},1}(m,T),
\end{aligned}$$

which holds $\forall x \in \mathcal{X}_{t,i}, t \in [T], i \in [N]$.

Also note that according to Lemma C.1 of Zhang et al. (2021), if conditions 3 and 4 on $m$ listed in App. C.1, as well as the condition on $\eta$ equation 2, are satisfied, then we have with probability of at least $1 - \delta_5$ over random initializations $\theta_0$ that $\left\|\theta_t^i - \theta_0\right\|_2 \leq 2\sqrt{t/m\lambda}, \forall i \in [N]$. An immediate implication is that the aggregated NN parameters $\theta_{\text{sync,NN}} = \frac{1}{N}\sum_{i=1}^{N}\theta_t^i$ also satisfies:

$$\left\|\theta_{\text{sync,NN}} - \theta_0\right\|_2 = \left\|\frac{1}{N}\sum_{i=1}^{N}\theta_t^i - \theta_0\right\|_2 \leq \frac{1}{N}\sum_{i=1}^{N}\left\|\theta_t^i - \theta_0\right\|_2 \leq 2\sqrt{t/m\lambda}.$$

This implies that equation 7 holds for $\theta_{\text{sync,NN}}$ with probability of at least $1 - 2\delta_5$:

$$|f(x;\theta_{\text{sync,NN}}) - \langle g(x;\theta_0), \theta_{\text{sync,NN}} - \theta_0 \rangle| \leq \varepsilon_{\text{linear},1}(m,T). \qquad (8)$$

Next, note that the $\theta_t^i$ is obtained by training only using agent $i$'s local observations (line 14 of Algo. 1). Define $\theta_{t,i}^{\text{local}} = (V_{t,i}^{\text{local}})^{-1}(\sum_{\tau=1}^{t}y_{\tau,i}g(x_{\tau,i};\theta_0)/\sqrt{m})$. Note that $\theta_{t,i}^{\text{local}}$ is calculated in the same way as $\overline{\theta}_{t,i}$ (line 4 of Algo. 1), except that its calculation only involves agent $i$'s local observations. Next, making use of Lemmas C.1 and C.4 of Zhang et al. (2021), we can follow similar

steps as equation C.3 of Zhang et al. (2021) (in Appendix C.2 of Zhang et al. (2021)) to show that there exists constants $C_2 > 0$ and $C_3 > 0$ such that we have $\forall x \in \mathcal{X}_{t,i}, t \in [T], i \in [N]$ that

$$
\begin{aligned}
|\langle g(x;\theta_0), &\theta_t^i - \theta_0\rangle - \langle g(x;\theta_0)/\sqrt{m}, \theta_{t,i}^{\text{local}}\rangle| \\
&\leq C_2(1-\eta m\lambda)^J\sqrt{tL/\lambda} + C_3 m^{-1/6}\sqrt{\log m}L^4 t^{5/3}\lambda^{-5/3}(1+\sqrt{t/\lambda}) \\
&\leq C_2(1-\eta m\lambda)^J\sqrt{TL/\lambda} + C_3 m^{-1/6}\sqrt{\log m}L^4 T^{5/3}\lambda^{-5/3}(1+\sqrt{T/\lambda}) \\
&\triangleq \varepsilon_{\eta,J} + \varepsilon_{\text{linear},2}(m,T).
\end{aligned}
\tag{9}
$$

We refer to $\langle g(x;\theta_0)/\sqrt{m}, \theta_{t,i}^{\text{local}}\rangle$ as the **linear prediction** because it is the prediction of the linear model with the neural tangent features $g(x;\theta_0)/\sqrt{m}$ as the input features, conditioned on the local observations of agent $i$. Note that similar to equation 8 which also relies on Lemma C.1 of Zhang et al. (2021), equation 9 also requires conditions 3 and 4 on $m$, as well as the condition on $\eta$, in App. C.1 to be satisfied. equation 9 holds with probability of at least $1 - 2\delta_5$, where the error probabilities come from the use of Lemmas C.1 and C.4 of Zhang et al. (2021).

Next, we can bound the difference between $f(x;\theta_{\text{sync,NN}})$ (i.e., the prediction of the NN with the aggregated parameters) and the averaged linear predictions of all agents calculated using their local observations:

$$
\begin{aligned}
|f(x;\theta_{\text{sync,NN}}) - \frac{1}{N}\sum_{i=1}^N \langle g(x;\theta_0)/\sqrt{m}, \theta_{t,i}^{\text{local}}\rangle| &\leq |f(x;\theta_{\text{sync,NN}}) - \frac{1}{N}\sum_{i=1}^N \langle g(x;\theta_0), \theta_t^i - \theta_0\rangle| \\
&+ |\frac{1}{N}\sum_{i=1}^N \langle g(x;\theta_0), \theta_t^i - \theta_0\rangle - \frac{1}{N}\sum_{i=1}^N \langle g(x;\theta_0)/\sqrt{m}, \theta_{t,i}^{\text{local}}\rangle| \\
&\leq |f(x;\theta_{\text{sync,NN}}) - \langle g(x;\theta_0), \theta_{\text{sync,NN}} - \theta_0\rangle| \\
&+ \frac{1}{N}\sum_{i=1}^N |\langle g(x;\theta_0), \theta_t^i - \theta_0\rangle - \langle g(x;\theta_0)/\sqrt{m}, \theta_{t,i}^{\text{local}}\rangle| \\
&\leq \varepsilon_{\text{linear},1}(m,T) + \frac{1}{N}\sum_{i=1}^N (\varepsilon_{\eta,J} + \varepsilon_{\text{linear},2}(m,T)) \\
&\leq \varepsilon_{\text{linear},1}(m,T) + \varepsilon_{\eta,J} + \varepsilon_{\text{linear},2}(m,T) \\
&\triangleq \varepsilon_{\text{linear}}(m,T).
\end{aligned}
\tag{10}
$$

In the second inequality, we plugged in the definition of $\theta_{\text{sync,NN}} = \frac{1}{N}\sum_{i=1}^N \theta_t^i$. In the third inequality, we have made use of equation 8 and equation 9; equation 10 holds with probability of at least $1 - 4\delta_5$, where the error probabilities come from equation 8 ($2\delta_5$) and equation 9 ($2\delta_5$), respectively. Now we replace $\delta_5$ by $\delta_5/4$, which ensures that equation 10 holds with probability of at least $1 - \delta_5$. This will only introduce a factor of 4 within the $\log$ of condition 3 on $m$ (App. C.1), which is ignored since it can be absorbed by the constant $C$.

**Step 2: Linear Prediction Is Close to the Reward Function $h(x)$**

In the proof in this section, we will also need a "local" variant of the confidence bound of Lemma 3, i.e., the confidence bound of Lemma 3 calculated only using the local observations of an agent $i$:

**Lemma 5** (Zhang et al. (2021)). *We have with probability of at least $1 - \delta_4$ for all $t \in \mathcal{T}_{-1} \subset [T], i \in [N]$, that*

$$
|h(x) - \langle g(x;\theta_0)/\sqrt{m}, \theta_{t,i}^{local}\rangle| \leq \nu_{TK}\sqrt{\lambda}\big\|g(x;\theta_0)/\sqrt{m}\big\|_{(V_{t,i}^{local})^{-1}}, \forall x \in \mathcal{X}_{t,i}.
$$

*Proof.* Similar to the proof of Lemma 3 (App. C.3.1), the proof of Lemma 5 also requires of Lemmas B.3, B.6 and B.7 from Zhang et al. (2021), which, in this case, can be directly applied to our setting (except that we need an additional union bound over all $N$ agents). The implication of the additional union bound on the error probabilities is taken care of by the additional term of $N$ within the $\log$ in the expression of $\nu_{TK}$ (Lemma 4), and in conditions 1 and 2 on $m$ (see App. C.1, and also Lemmas 2 and 1). The required lower bounds on $m$ by the local variants of Lemmas B.3 and B.7 (required in

the proof here) are smaller than those given in Lemmas 2 and 1 and hence do not need to appear in the conditions in Appendix C.1. By letting the sum of the three error probabilities (resulting from the applications of Lemmas B.3, B.6 and B.7 of Zhang et al. (2021)) be $\delta_4$, we can ensure that Lemma 5 holds with probability of at least $1 - \delta_4$. For simplicity, we let the error probability for Lemma B.6 be $\delta_4$, which leads to the cleaner expression of $\nu_{TK}$ in Lemma 4. This means that the error probabilities for Lemmas B.3 and B.7 are very small, which can be accounted for by simply increasing the value of the absolute constant $C$ in conditions 1 and 2 on $m$ (App. C.1) and hence does not affect our main theoretical analysis. □

**Step 3: Combining Results from Step 1 and Step 2**

Next, we are ready to prove the validity of $\text{UCB}_{t,i}^b$ by using the averaged linear prediction $\frac{1}{N} \sum_{i=1}^{N} \langle g(x; \theta_0)/\sqrt{m}, \theta_{t,i}^{\text{local}} \rangle$ as an intermediate term:

$$
\begin{aligned}
&|f(x; \theta_{\text{sync,NN}}) - h(x)| \\
&= \left| f(x; \theta_{\text{sync,NN}}) - \frac{1}{N} \sum_{i=1}^{N} \langle g(x; \theta_0)/\sqrt{m}, \theta_{t,i}^{\text{local}} \rangle + \frac{1}{N} \sum_{i=1}^{N} \langle g(x; \theta_0)/\sqrt{m}, \theta_{t,i}^{\text{local}} \rangle - h(x) \right| \\
&\leq \frac{1}{N} \sum_{i=1}^{N} |\langle g(x; \theta_0)/\sqrt{m}, \theta_{t,i}^{\text{local}} \rangle - h(x)| + \varepsilon_{\text{linear}}(m, T) \\
&\leq \frac{1}{N} \sum_{i=1}^{N} \nu_{TK} \sqrt{\lambda} \|g(x; \theta_0)/\sqrt{m}\|_{(V_{t,i}^{\text{local}})^{-1}} + \varepsilon_{\text{linear}}(m, T) \\
&= \frac{1}{N} \sum_{i=1}^{N} \nu_{TK} \sqrt{\lambda} \|g(x; \theta_0)/\sqrt{m}\|_{(V_i^{\text{local}})^{-1}} + \varepsilon_{\text{linear}}(m, T)
\end{aligned}
$$

(11)

The second inequality has made use of equation 10, and the third inequality follows from Lemma 5. In the last inequality, we have made the substitution of $(V_i^{\text{local}})^{-1} = (V_{t,i}^{\text{local}})^{-1}$. This is because in the proof of Lemma 4 here, we only consider the iterations of $t \in \mathcal{T}_{-1} \triangleq \{t_p - 1\}_{p=2,\dots,P-1}$, i.e., *the last iteration of every epoch*. As a result, this ensures that $(V_i^{\text{local}})^{-1} = (V_{t,i}^{\text{local}})^{-1}$ because every time the central server obtains $(V_i^{\text{local}})^{-1}$ through $(V_i^{\text{local}})^{-1} = (V_{t,i}^{\text{local}})^{-1}, \forall i \in [N]$ (line 4 of Algo. 2), we have that the current iteration $t$ is *the last iteration of the previous epoch*. As a results, equation 11 holds with probability of at least $1 - \delta_4 - \delta_5$, in which the error probabilities come from equation 10 ($\delta_5$) and Lemma 5 ($\delta_4$). In other words, Lemma 4 (i.e., the validity of $\text{UCB}_{t,i}^b$) holds with probability of at least $1 - \delta_4 - \delta_5$.

□

## C.4 REGRET UPPER BOUND FOR GOOD EPOCHS

In this section, we derive an upper bound on the total regrets incurred in all good epochs $\mathcal{E}^{\text{good}}$ (defined in App. C.2).

### C.4.1 AUXILIARY INEQUALITY

We firstly derive an auxiliary result which will be used in the proofs later.

For agent $i$ and iteration $t$ in a good epoch $p \in \mathcal{E}^{\text{good}}$, we have that

$$
\begin{aligned}
\sqrt{\lambda}\big\|g(x;\theta_0)/\sqrt{m}\big\|_{\overline{V}_{t,i}^{-1}} &= \sqrt{\lambda g(x;\theta_0)^\top \overline{V}_{t,i}^{-1} g(x;\theta_0)/m} \\
&\leq \sqrt{\lambda g(x;\theta_0)^\top \widetilde{V}_{t,i}^{-1} g(x;\theta_0)/m \frac{\det\widetilde{V}_{t,i}}{\det\overline{V}_{t,i}}} \\
&\leq \sqrt{\lambda g(x;\theta_0)^\top \widetilde{V}_{t,i}^{-1} g(x;\theta_0)/m \frac{\det V_p}{\det V_{p-1}}} \\
&\leq \sqrt{e\lambda g(x;\theta_0)^\top \widetilde{V}_{t,i}^{-1} g(x;\theta_0)/m} \\
&= \sqrt{e\lambda}\big\|g(x;\theta_0)/\sqrt{m}\big\|_{\widetilde{V}_{t,i}^{-1}} .
\end{aligned}
\tag{12}
$$

Recall that $\overline{V}_{t,i}$ (line 4 of Algo. 1) is used by agent $i$ in iteration $t$ to select $x_{t,i}$ (via $\text{UCB}_{t,i}^a$), and that the matrix $\widetilde{V}_{t,i}$ is defined for the hypothetical agent which sequentially chooses all $TN$ queries $\{x_{t,i}\}_{t\in[T],i\in[N]}$ in a round-robin fashion (App. C.2). The first inequality in equation 12 above follows from Lemma 12 of Abbasi-Yadkori et al. (2011). The second inequality is because $V_p$ contains more information than $\widetilde{V}_{t,i}$ (since $V_p$ is calculated using all the inputs selected *after* epoch $p$), and $V_{p-1}$ contains less information than $\overline{V}_{t,i}$ (because compared with $V_{p-1}$, $\overline{V}_{t,i}$ additionally contains the local inputs selected by agent $i$ in the current epoch $p$). In the last inequality, we have made use of the definition of good epochs, i.e., $(\det V_p)/(\det V_{p-1}) \leq e$ (App. C.2).

### C.4.2 Upper Bound on the Instantaneous Regret $r_{t,i}$

Here we assume that both $\text{UCB}_{t,i}^a$ and $\text{UCB}_{t,i}^b$ hold (hence we ignore the error probabilities here), which we have proved in App. C.3. We now derive an upper bound on the instantaneous regret $r_{t,i} = h(x_{t,i}^*) - h(x_{t,i})$ for agent $i$ and iteration $t$ in a good epoch $p \in \mathcal{E}^{\text{good}}$:

$$
\begin{aligned}
r_{t,i} &= h(x_{t,i}^*) - h(x_{t,i}) \\
&= \alpha h(x_{t,i}^*) + (1-\alpha)h(x_{t,i}^*) - h(x_{t,i}) \\
&\leq \alpha\text{UCB}_{t,i}^b(x_{t,i}^*) + \alpha\varepsilon_{\text{linear}}(m,T) + (1-\alpha)\text{UCB}_{t,i}^a(x_{t,i}^*) - h(x_{t,i}) \\
&\leq \alpha\text{UCB}_{t,i}^b(x_{t,i}) + (1-\alpha)\text{UCB}_{t,i}^a(x_{t,i}) + \alpha\varepsilon_{\text{linear}}(m,T) - h(x_{t,i}) \\
&= \alpha\left(\text{UCB}_{t,i}^b(x_{t,i}) - h(x_{t,i})\right) + (1-\alpha)\left(\text{UCB}_{t,i}^a(x_{t,i}) - h(x_{t,i})\right) + \alpha\varepsilon_{\text{linear}}(m,T) \\
&\leq \alpha\left(2\nu_{TK}\frac{1}{N}\sum_{j=1}^{N}\sqrt{\lambda}\big\|g(x_{t,i};\theta_0)/\sqrt{m}\big\|_{(V_j^{\text{local}})^{-1}} + \varepsilon_{\text{linear}}(m,T)\right) + \\
&\quad (1-\alpha)\left(2\nu_{TKN}\sqrt{\lambda}\big\|g(x_{t,i};\theta_0)/\sqrt{m}\big\|_{\overline{V}_{t,i}^{-1}}\right) + \alpha\varepsilon_{\text{linear}}(m,T) \\
&\leq \alpha\left(2\nu_{TK}\frac{1}{N}\sum_{j=1}^{N}\sqrt{\lambda}\big\|g(x_{t,i};\theta_0)/\sqrt{m}\big\|_{(V_j^{\text{local}})^{-1}} + \varepsilon_{\text{linear}}(m,T)\right) + \\
&\quad (1-\alpha)\left(2\nu_{TKN}\sqrt{e\lambda}\big\|g(x_{t,i};\theta_0)/\sqrt{m}\big\|_{\widetilde{V}_{t,i}^{-1}}\right) + \alpha\varepsilon_{\text{linear}}(m,T) \\
&= \alpha 2\nu_{TK}\frac{1}{N}\sum_{j=1}^{N}\sqrt{\lambda}\big\|g(x_{t,i};\theta_0)/\sqrt{m}\big\|_{(V_j^{\text{local}})^{-1}} + \\
&\quad (1-\alpha)2\nu_{TKN}\sqrt{e\lambda}\big\|g(x_{t,i};\theta_0)/\sqrt{m}\big\|_{\widetilde{V}_{t,i}^{-1}} + 2\alpha\varepsilon_{\text{linear}}(m,T) \\
&\triangleq (1-\alpha)2\nu_{TKN}\sqrt{e}\widetilde{\sigma}_{t,i}(x_{t,i}) + \alpha 2\nu_{TK}\frac{1}{N}\sum_{j=1}^{N}\widetilde{\sigma}_{t_p-1,j}^{\text{local}}(x_{t,i}) + 2\alpha\varepsilon_{\text{linear}}(m,T).
\end{aligned}
\tag{13}
$$

The first inequality makes use of Lemma 3 (i.e., the validity of $\text{UCB}_{t,i}^a$) and Lemma 4 (i.e., the validity of $\text{UCB}_{t,i}^b$). The second inequality follows from the way in which $x_{t,i}$ is selected

(line 7 of Algo. 1): $x_{t,i} = \arg\max_{x \in \mathcal{X}_{t,i}} (1 - \alpha)\text{UCB}_{t,i}^a(x) + \alpha\text{UCB}_{t,i}^b(x)$. The third inequality again makes use of Lemma 3 and Lemma 4, as well as the expressions of $\text{UCB}_{t,i}^a$ and $\text{UCB}_{t,i}^b$. In the fourth inequality, we have made used of the auxiliary inequality of equation 12 we derived in the last section. Recall that we have discussed that $(V_i^{\text{local}})^{-1} = (V_{t,i}^{\text{local}})^{-1}$ for all $t = t_p - 1$ at the end of App. C.3.2. Therefore, in the last step, we have defined $\widetilde{\sigma}_{t_p-1,j}^{\text{local}}(x_{t,i}) \triangleq \sqrt{\lambda}\big\|g(x_{t,i};\theta_0)/\sqrt{m}\big\|_{(V_{t_p-1,j}^{\text{local}})^{-1}} = \sqrt{\lambda}\big\|g(x_{t,i};\theta_0)/\sqrt{m}\big\|_{(V_j^{\text{local}})^{-1}}$ which represents the GP posterior standard deviation (using the kernel of $\widetilde{k}(x,x') = g(x;\theta_0)^\top g(x';\theta_0)/m$) conditioned on all agent $j$'s local observations before iteration $t_p$. Note that $\widetilde{\sigma}_{t_p-1,j}^{\text{local}}(x_{t,i})$ is the same as the one defined in Sec. 3.3 of the main text where we explain the weight between the two UCBs. Similarly, we have also defined $\widetilde{\sigma}_{t,i}(x_{t,i}) \triangleq \sqrt{\lambda}\big\|g(x_{t,i};\theta_0)/\sqrt{m}\big\|_{\widetilde{V}_{t,i}^{-1}}$, which represents the GP posterior standard deviation conditioned on the observations of the hypothetical agent before $x_{t,i}$ is selected (App. C.2).

Next, we will separately derive upper bounds on the summation (across all good epochs and all agents) of the first and second terms of the upper bound from equation 13.

### C.4.3 UPPER BOUND ON THE SUM OF THE FIRST TERM OF EQUATION 13

Here, similar to Kassraie & Krause (2022), we denote as $\kappa_0$ an upper bound on the value of the NTK function for any input: $\langle g(x;\theta_0)/\sqrt{m}, g(x;\theta_0)/\sqrt{m}\rangle \leq \kappa_0, \forall x \in \mathcal{X}_{t,i}, t \in [T], i \in [N]$. As a result, we can use it to show that both $\widetilde{\sigma}_{t,i}(x)$ and $\widetilde{\sigma}_{t_p-1,j}^{\text{local}}(x)$ can be upper-bounded: $\widetilde{\sigma}_{t,i}(x) \leq \sqrt{\kappa_0}$ and $\widetilde{\sigma}_{t_p-1,j}^{\text{local}}(x) \leq \sqrt{\kappa_0}$. To show this, following the notations of Appendix C.2, we denote $\widetilde{V}_{t,i} = J_{t,i}J_{t,i}^\top + \lambda I$ where $J_{t,i} = \left[\big[g(x_{\tau,j};\theta_0)\big]_{\tau \in [t-1], j \in [N]}, \big[g(x_{t,j};\theta_0)\big]_{j \in [i]}\right]$ which is a $p_0 \times [(t-1)N + i]$ matrix. Then we have

$$
\begin{aligned}
\widetilde{\sigma}_{t,i}^2(x) &= \lambda\big\|g(x;\theta_0)/\sqrt{m}\big\|_{\widetilde{V}_{t,i}^{-1}}^2 \\
&= \lambda g(x;\theta_0)^\top (J_{t,i}J_{t,i}^\top + \lambda I)^{-1} g(x;\theta_0)/m \\
&= \lambda g(x;\theta_0)^\top \Big(\frac{1}{\lambda}I - \frac{1}{\lambda}J_{t,i}\big(I + J_{t,i}^\top \frac{1}{\lambda}J_{t,i}\big)^{-1}J_{t,i}^\top \frac{1}{\lambda}\Big)g(x;\theta_0)/m \\
&= g(x;\theta_0)^\top g(x;\theta_0)/m - (g(x;\theta_0)^\top/\sqrt{m})J_{t,i}\big(\lambda I + J_{t,i}^\top J_{t,i}\big)^{-1}J_{t,i}^\top(g(x;\theta_0)/\sqrt{m}) \\
&= g(x;\theta_0)^\top g(x;\theta_0)/m - \big\|(g(x;\theta_0)^\top/\sqrt{m})J_{t,i}\big\|_{\left(\lambda I + J_{t,i}^\top J_{t,i}\right)^{-1}}^2 \\
&\leq g(x;\theta_0)^\top g(x;\theta_0)/m \leq \kappa_0
\end{aligned}
\tag{14}
$$

where we used the matrix inversion lemma in the third equality. Using similar derivations also allows us to show that $(\widetilde{\sigma}_{t_p-1,j}^{\text{local}}(x))^2 \leq \kappa_0$. Therefore, we have that $\widetilde{\sigma}_{t,i}(x) \leq \sqrt{\kappa_0}$ and $\widetilde{\sigma}_{t_p-1,j}^{\text{local}}(x) \leq \sqrt{\kappa_0}$.

Denoting the set of iterations from all good epochs as $\mathcal{T}^{\text{good}}$, we can derive an upper bound the first term of equation 13, summed across all agents $i \in [N]$ and all iteration in good epochs $\mathcal{T}^{\text{good}}$:

$$
\begin{aligned}
\sum_{i=1}^{N} \sum_{t \in \mathcal{T}^{\text{good}}} (1-\alpha) 2\nu_{TKN} \sqrt{e} \widetilde{\sigma}_{t,i}(x_{t,i}) &\overset{(a)}{\leq} 2\sqrt{e}\nu_{TKN} \sum_{i=1}^{N} \sum_{t=1}^{T} \widetilde{\sigma}_{t,i}(x_{t,i}) \\
&\overset{(b)}{=} 2\sqrt{e}\nu_{TKN} \sum_{i=1}^{N} \sum_{t=1}^{T} \min\{\widetilde{\sigma}_{t,i}(x_{t,i}), \sqrt{\kappa_0}\} \\
&\overset{(c)}{\leq} 2\sqrt{e}\nu_{TKN} \sum_{i=1}^{N} \sum_{t=1}^{T} \min\{\sqrt{\kappa_0}\widetilde{\sigma}_{t,i}(x_{t,i}), \sqrt{\kappa_0}\} \\
&\leq 2\sqrt{e}\nu_{TKN}\sqrt{\kappa_0} \sum_{i=1}^{N} \sum_{t=1}^{T} \min\{\widetilde{\sigma}_{t,i}(x_{t,i}), 1\} \\
&\overset{(d)}{\leq} 2\sqrt{e}\nu_{TKN}\sqrt{\kappa_0} \sqrt{TN \sum_{i=1}^{N} \sum_{t=1}^{T} \min\{\widetilde{\sigma}_{t,i}^2(x_{t,i}), 1\}} \\
&\overset{(e)}{\leq} 2\sqrt{e}\nu_{TKN}\sqrt{\kappa_0} \sqrt{TN[2\lambda \log\det(\lambda^{-1}\mathbf{K}_{TN} + I)]} \\
&\overset{(f)}{\leq} 2\sqrt{2e}\nu_{TKN}\sqrt{\kappa_0} \sqrt{TN\lambda[\log\det(\lambda^{-1}\mathbf{H} + I) + 1]} \\
&= 2\sqrt{e}\nu_{TKN}\sqrt{\kappa_0} \sqrt{TN\lambda\left[\widetilde{d}\log(1 + TNK/\lambda) + 1\right]}
\end{aligned}
\tag{15}
$$

Step $(a)$ follows from $\alpha \leq 1, \forall t \geq 1$ and summing across all iterations $[T]$ instead of only those iterations $\mathcal{T}^{\text{good}}$ in good epochs. Step $(b)$ follows because $\widetilde{\sigma}_{t,i}(x) \leq \sqrt{\kappa_0}$ as discussed above. In step $(c)$, we have assumed that $\kappa_0 \geq 1$; however, if $\kappa_0 < 1$, the proof still goes through since we can directly upper-bound $\min\{\widetilde{\sigma}_{t,i}(x_{t,i}), \sqrt{\kappa_0}\}$ by $\min\{\widetilde{\sigma}_{t,i}(x_{t,i}), 1\}$, after which the only modification we need to make to the equation above is to remove the dependency on multiplicative term of $\sqrt{\kappa_0}$. Step $(d)$ results from the Cauchy–Schwarz inequality. Step $(e)$ can be derived following the proof of Lemma 4.8 of Zhang et al. (2021) (in Appendix B.7 of Zhang et al. (2021)). Step $(f)$ follows from Lemma 1 and hence holds with probability of at least $1 - \delta_1$. The last equality simply plugs in the definition of the effective dimension $\widetilde{d}$ (Sec. 2).

### C.4.4 Upper Bound on the Sum of the Second term of equation 13

In this subsection, we derive an upper bound on the sum of the second term in equation equation 13 across all good epochs and all agents.

For the proof here, we need a "local" version of Lemma 1, i.e., a version of Lemma 1 which only makes use of the contexts of an agent $i$. Define $\mathbf{K}_{t,i}$ as the local counterpart to $\mathbf{K}_{t'}$ (from Lemma 1), i.e., $\mathbf{K}_{t,i}$ is the $t \times t$ matrix calculated using only agent $j$'s local contexts up to (and including) iteration $t$. Specifically, define $J_{t,i} \triangleq [g(x_{\tau,i}; \theta_0)]_{\tau \in [t]}$ which is a $p_0 \times t$ matrix, then $\mathbf{K}_{t,i}$ is defined as $\mathbf{K}_{t,i} \triangleq J_{t,i}^{\top} J_{t,i}/m$, which is a $t \times t$ matrix. Also recall that in the main text, we have defined $\mathbf{H}_i$ as the local counterpart of $\mathbf{H}$ for agent $i$ (Sec. 2). The next lemma gives our desired local version of Lemma 1.

**Lemma 6** (Lemma B.7 of Zhang et al. (2021)). *If $m \geq CT^6 K^6 L^6 \log(TNKL/\delta_6)$, we have with probability of at least $1 - \delta_6$ that*

$$\log\det(I + \lambda^{-1}\mathbf{K}_{t,i}) \leq \log\det(I + \lambda^{-1}\mathbf{H}_i) + 1,$$

*for all $t \in [T], i \in [N]$.*

We needed to take a union bound over all $N$ agents, which explains the factor of $N$ within the $\log$ in the lower bound on $m$ given in Lemma 6. Note that the required lower bound on $m$ by Lemma 6 is smaller than that of Lemma 1 (by a factor of $N^6$), therefore, the condition on $m$ in Lemma 6 is ignored in the conditions listed in App. C.1.

Of note, throughout the entire epoch $p$, $\widetilde{\sigma}^{\text{local}}_{t_p-1,j}(x_{t,i})$ is calculated *conditioned on all the local observations of agent $j$ before iteration $t_p$*. Denote by $\mathcal{T}^{(p)}$ the iteration indices in epoch $p$: $\mathcal{T}^{(p)} = \{t_p, \ldots, t_p + E_p - 1\}$. In the proof in this section, as we have discussed in the first paragraph of Sec. 4.1, we analyze a simpler variant of our algorithm where we only set $\alpha > 0$ in the first iteration after a communication round, i.e., $\alpha > 0, \forall t \in \{t_p\}_{p \in [P]}$ and $\alpha = 0, \forall t \in [T] \setminus \{t_p\}_{p \in [P]}$. Now we are ready to derive an upper bound on the second term in equation 13, summed over all agents and all good epochs:

$$
\sum_{i=1}^{N} \sum_{p \in \mathcal{E}^{\text{good}}} \sum_{t \in \mathcal{T}^{(p)}} \alpha 2\nu_{TK} \frac{1}{N} \sum_{j=1}^{N} \widetilde{\sigma}^{\text{local}}_{t_p-1,j}(x_{t,i}) \overset{(a)}{\leq} 2\nu_{TK} \frac{1}{N} \sum_{i=1}^{N} \sum_{p \in [P]} \sum_{t \in \mathcal{T}^{(p)}} \alpha \sum_{j=1}^{N} \widetilde{\sigma}^{\text{local}}_{t_p-1,j}(x_{t,i})
$$

$$
\overset{(b)}{\leq} 2\nu_{TK} \frac{1}{N} \sum_{i=1}^{N} \sum_{j=1}^{N} \sum_{p \in [P]} \alpha \widetilde{\sigma}^{\text{local}}_{t_p-1,j}(x_{t_p,i})
$$

$$
\overset{(c)}{\leq} 2\nu_{TK} \frac{1}{N} \sum_{i=1}^{N} \sum_{j=1}^{N} \sum_{p \in [P]} \widetilde{\sigma}^{\text{local}}_{t_p-1,j}(x_{t_p,j})
$$

$$
\overset{(d)}{\leq} 2\nu_{TK} \frac{1}{N} \sum_{i=1}^{N} \sum_{j=1}^{N} \sum_{t=1}^{T} \widetilde{\sigma}^{\text{local}}_{t-1,j}(x_{t,j})
$$

$$(16)$$

The inequality in step $(a)$ results from summing across all epoch $p \in [P]$ instead of only good epochs $p \in \mathcal{E}^{\text{good}}$. Step $(b)$ follows since $\alpha_t = 0, \forall t \in [T] \setminus \{t_p\}_{p \in [P]}$ as we discussed above, therefore, for every epoch $p$, we only need to keep the first term of $t = t_p$ in the summation of $t \in \mathcal{T}^{(p)}$. To understand step $(c)$, recall that in the main text (Sec. 3.3), we have defined: $\widetilde{\sigma}^{\text{local}}_{t,i,\min} \triangleq \min_{x \in \mathcal{X}} \widetilde{\sigma}^{\text{local}}_{t,i}(x)$ and $\widetilde{\sigma}^{\text{local}}_{t,i,\max} \triangleq \max_{x \in \mathcal{X}} \widetilde{\sigma}^{\text{local}}_{t,i}(x), \forall i \in [N]$. Next, note that our algorithm selects $\alpha$ by: $\alpha = \min_{i \in [N]} \alpha_{t,i}$ (line 4 of Algo. 2) where $\alpha_{t,i} = \widetilde{\sigma}^{\text{local}}_{t,i,\min}/\widetilde{\sigma}^{\text{local}}_{t,i,\max}$ (line 15 of Algo. 1) and $t = t_p - 1$ since $\alpha_{t,i}$ is calculated only in the last iteration of every epoch. As a result, we have that

$$
\alpha = \min_{i \in [N]} \alpha_{t_p-1,i} = \min_{i \in [N]} \frac{\widetilde{\sigma}^{\text{local}}_{t_p-1,i,\min}}{\widetilde{\sigma}^{\text{local}}_{t_p-1,i,\max}} \leq \frac{\widetilde{\sigma}^{\text{local}}_{t_p-1,j,\min}}{\widetilde{\sigma}^{\text{local}}_{t_p-1,j,\max}} \leq \frac{\widetilde{\sigma}^{\text{local}}_{t_p-1,j}(x_{t_p,j})}{\widetilde{\sigma}^{\text{local}}_{t_p-1,j}(x_{t_p,i})},
$$

which tells us that $\alpha \widetilde{\sigma}^{\text{local}}_{t_p-1,j}(x_{t_p,i}) \leq \widetilde{\sigma}^{\text{local}}_{t_p-1,j}(x_{t_p,j})$ and hence leads to step $(c)$. Step $(d)$ results from summing across all iterations $[T]$ instead of only the first iteration of every epoch.

Next, we can derive an upper bound on the inner summation over $t = 1, \ldots, T$ from equation 16:

$$
\sum_{t=1}^{T} \widetilde{\sigma}^{\text{local}}_{t-1,j}(x_{t,j}) \overset{(a)}{\leq} \sqrt{\kappa_0} \sum_{t=1}^{T} \min\{\widetilde{\sigma}^{\text{local}}_{t-1,j}(x_{t,j}), 1\}
$$

$$
\overset{(b)}{\leq} \sqrt{\kappa_0} \sqrt{T \sum_{t=1}^{T} \min\{\left(\widetilde{\sigma}^{\text{local}}_{t-1,j}(x_{t,j})\right)^2, 1\}}
$$

$$
\overset{(c)}{\leq} \sqrt{\kappa_0} \sqrt{T[2\lambda \log \det(\lambda^{-1}\mathbf{K}_{T,j} + I)]}
$$

$$
\overset{(d)}{\leq} \sqrt{2}\sqrt{\kappa_0} \sqrt{T\lambda[\log \det(\lambda^{-1}\mathbf{H}_j + I) + 1]}
$$

$$
= \sqrt{2}\sqrt{\kappa_0} \sqrt{T\lambda \left[\widetilde{d}_j \log(1 + TK/\lambda)) + 1\right]}.
$$

$$(17)$$

Step $(a)$ is obtained in the same way as steps $(b)$ and $(c)$ in equation 15 (App. C.4.3), i.e., we have made use of $\widetilde{\sigma}^{\text{local}}_{t_p-1,j}(x) \leq \sqrt{\kappa_0}$ and assumed that $\kappa_0 \geq 1$. Again note that if $\kappa_0 < 1$, then the proof still goes through since $\widetilde{\sigma}^{\text{local}}_{t-1,j}(x_{t,j}) \leq \min\{\widetilde{\sigma}^{\text{local}}_{t-1,j}(x_{t,j}), \sqrt{\kappa_0}\} \leq \min\{\widetilde{\sigma}^{\text{local}}_{t-1,j}(x_{t,j}), 1\}$, after which the only modification we need to make to the equation above is to remove the dependency on multiplicative term of $\sqrt{\kappa_0}$. Step $(b)$ makes use of the Cauchy–Schwarz inequality. Step $(c)$,

similar to step $(e)$ of equation 15, is derived following the proof of Lemma 4.8 of Zhang et al. (2021) (in Appendix B.7 of Zhang et al. (2021)). Step $(d)$ follows from Lemma 6 and hence holds with probability of at least $1 - \delta_6$. In the last equality, we have simply plugged in the definition of $\widetilde{d}_j$ (Sec. 2).

Now we can plug equation 17 into equation 16 to obtain

$$
\sum_{i=1}^{N} \sum_{p \in \mathcal{E}^{\text{good}}} \sum_{t \in \mathcal{T}^{(p)}} \alpha 2\nu_{TK} \frac{1}{N} \sum_{j=1}^{N} \widetilde{\sigma}_{t_p-1,j}^{\text{local}}(x_{t,i})
$$
$$
\leq 2\nu_{TK} \frac{1}{N} \sum_{i=1}^{N} \sum_{j=1}^{N} \sqrt{2}\sqrt{\kappa_0} \sqrt{T\lambda \left[ \widetilde{d}_j \log(1 + TK/\lambda)) + 1 \right]} \tag{18}
$$
$$
= 2\sqrt{2}\nu_{TK}\sqrt{\kappa_0} \sum_{j=1}^{N} \sqrt{T\lambda \left[ \widetilde{d}_j \log(1 + TK/\lambda)) + 1 \right]}.
$$

### C.4.5 Putting Things Together

Finally, recall that our derived upper bound on $r_{t,i}$ in equation 13 contains three terms (the third term is simply an error term), and now we can make use of our derived upper bound on the first term (App. C.4.3) and the second term (App. C.4.4), summed over all agents and all good epochs, to obtain an upper bound on the total regrets incurred in all good epochs:

$$
R_T^{\text{good}} = \sum_{i=1}^{N} \sum_{t \in \mathcal{T}^{\text{good}}} r_{t,i}
$$
$$
\leq 2\sqrt{e}\nu_{TKN}\sqrt{\kappa_0}\sqrt{TN\lambda \left[ \widetilde{d} \log(1 + TNK/\lambda) + 1 \right]} +
$$
$$
2\sqrt{2}\nu_{TK}\sqrt{\kappa_0} \sum_{j=1}^{N} \sqrt{T\lambda \left[ \widetilde{d}_j \log(1 + TK/\lambda)) + 1 \right]} + TN\varepsilon_{\text{linear}}(m, T) \tag{19}
$$
$$
= \widetilde{O}\left( \sqrt{\widetilde{d}}\sqrt{TN\widetilde{d}} + \sqrt{\widetilde{d}_{\max}}N\sqrt{T\widetilde{d}_{\max}} + TN\varepsilon_{\text{linear}}(m, T) \right)
$$
$$
= \widetilde{O}\left( \widetilde{d}\sqrt{TN} + \widetilde{d}_{\max}N\sqrt{T} + TN\varepsilon_{\text{linear}}(m, T) \right).
$$

In the second last equality, we have used $\nu_{TKN} = \widetilde{O}(\sqrt{\widetilde{d}})$ and $\nu_{TK} = \widetilde{O}(\sqrt{\widetilde{d}_{\max}})$.

## C.5 REGRET UPPER BOUND FOR BAD EPOCHS

In this section, we derive an upper bound on the total regrets from all bad epochs. To begin with, we firstly derive an upper bound on the total regrets of any bad epoch $p$ denoted as $R^{[p]}$:

$$
\begin{aligned}
R^{[p]} = \sum_{i=1}^{N} \sum_{t=t_p}^{t_p+E_p-1} r_{t,i} &\overset{(a)}{\leq} \sum_{i=1}^{N} \left( 2 + 2 + \sum_{t=t_p+1}^{t_p+E_p-2} r_{t,i} \right) \\
&\overset{(b)}{\leq} \sum_{i=1}^{N} \left[ 4 + \sum_{t=t_p+1}^{t_p+E_p-2} \left( \mathrm{UCB}_{t,i}^a(x_{t,i}^*) - h(x_{t,i}) \right) \right] \\
&\overset{(c)}{\leq} \sum_{i=1}^{N} \left[ 4 + \sum_{t=t_p+1}^{t_p+E_p-2} \left( \mathrm{UCB}_{t,i}^a(x_{t,i}) - h(x_{t,i}) \right) \right] \\
&\overset{(d)}{\leq} \sum_{i=1}^{N} \left[ 4 + \sum_{t=t_p+1}^{t_p+E_p-2} 2\nu_{TKN}\sqrt{\lambda} \left\| g(x_{t,i};\theta_0)/\sqrt{m} \right\|_{\overline{V}_{t,i}^{-1}} \right] \\
&\overset{(e)}{\leq} \sum_{i=1}^{N} \left( 4 + 2\nu_{TKN}\sqrt{\kappa_0} \sum_{t=t_p}^{t_p+E_p-2} \min\{\sqrt{\lambda}\left\| g(x_{t,i};\theta_0)/\sqrt{m} \right\|_{\overline{V}_{t,i}^{-1}}, 1\} \right) \\
&\overset{(f)}{\leq} \sum_{i=1}^{N} \left( 4 + 2\nu_{TKN}\sqrt{\kappa_0\lambda} \sum_{t=t_p}^{t_p+E_p-2} \min\{\left\| g(x_{t,i};\theta_0)/\sqrt{m} \right\|_{\overline{V}_{t,i}^{-1}}, 1\} \right)
\end{aligned}
\tag{20}
$$

Step $(a)$ follows from simply upper-bounding the regrets of the first and last iteration within this epoch by 2. Step $(b)$ makes use of the validity of $\mathrm{UCB}_{t,i}^a$ (Lemma 3). Step $(c)$ follows because $\alpha = 0, \forall t \in [T] \setminus \{t_p\}_{p \in [P]}$ (i.e., we set $\alpha = 0$ except for the first iteration of all epochs), which implies that after the first iteration of an epoch, $x_{t,i}$ is selected by only maximizing $\mathrm{UCB}_{t,i}^a$ (line 7 of Algo. 1). Step $(d)$ again uses Lemma 3, as well as the expression of $\mathrm{UCB}_{t,i}^a$. Step $(e)$ is obtained in the same way as steps $(b)$ and $(c)$ in equation 15 (App. C.4.3). Specifically, since $\langle g(x;\theta_0), g(x;\theta_0) \rangle \leq \kappa_0$ (App. C.4.3), therefore, $\sqrt{\lambda}\left\| g(x_{t,i};\theta_0)/\sqrt{m} \right\|_{\overline{V}_{t,i}^{-1}} \leq \sqrt{\kappa_0}$, which can be proved by following the same steps as equation 14. As a result, if we assume that $\kappa \geq 1$, then $\sqrt{\lambda}\left\| g(x_{t,i};\theta_0)/\sqrt{m} \right\|_{\overline{V}_{t,i}^{-1}} = \min\{\sqrt{\lambda}\left\| g(x_{t,i};\theta_0)/\sqrt{m} \right\|_{\overline{V}_{t,i}^{-1}}, \sqrt{\kappa_0}\} \leq \sqrt{\kappa_0} \min\{\sqrt{\lambda}\left\| g(x_{t,i};\theta_0)/\sqrt{m} \right\|_{\overline{V}_{t,i}^{-1}}, 1\}$; in the other case where $\kappa_0 < 1$, then $\sqrt{\lambda}\left\| g(x_{t,i};\theta_0)/\sqrt{m} \right\|_{\overline{V}_{t,i}^{-1}} = \min\{\sqrt{\lambda}\left\| g(x_{t,i};\theta_0)/\sqrt{m} \right\|_{\overline{V}_{t,i}^{-1}}, \sqrt{\kappa_0}\} \leq \min\{\sqrt{\lambda}\left\| g(x_{t,i};\theta_0)/\sqrt{m} \right\|_{\overline{V}_{t,i}^{-1}}, 1\}$. Here we have assumed $\kappa_0 \geq 1$ for simplicity, since when $\kappa_0 < 1$, the equation above still holds except that we can remove the dependency on $\sqrt{\kappa_0}$. Step $(f)$ follows because $\lambda = 1 + 2/T > 1$.

Next, we derive an upper bound on the inner summation in equation 20.

$$
\begin{aligned}
\sum_{t=t_p}^{t_p+E_p-2} &\min\{\left\| g(x_{t,i};\theta_0)/\sqrt{m} \right\|_{\overline{V}_{t,i}^{-1}}, 1\} \\
&\overset{(a)}{\leq} \sqrt{(E_p - 1) \sum_{t=t_p}^{t_p+E_p-2} \min\{\left\| g(x_{t,i};\theta_0)/\sqrt{m} \right\|_{\overline{V}_{t,i}^{-1}}^2, 1\}} \\
&\overset{(b)}{\leq} \sqrt{(E_p - 1) 2 \log \frac{\det \overline{V}_{t_p+E_p-2,i}}{\det \overline{V}_{t_p,i}}} \\
&\overset{(c)}{\leq} \sqrt{2((t_p + E_p - 2) - t_{\text{last}}) \log \frac{\det V_{t_p+E_p-2,i}}{\det V_{\text{last}}}} \\
&\overset{(d)}{\leq} \sqrt{2D}.
\end{aligned}
\tag{21}
$$

Step $(a)$ follows from the Cauchy–Schwarz inequality. Step $(b)$ makes use of Lemma 11 of Abbasi-Yadkori et al. (2011). In step $(c)$, we used the notations of $t_{\text{last}} = t_p - 1$, $\overline{V}_{t_p,i} = V_{\text{last}}$ (this is because in the first iteration $t_p$ of an epoch, $W_{\text{new},i} = \mathbf{0}_{p_0 \times p_0}$ and hence $\overline{V}_{t_p,i} = V_{\text{last}} = W_{\text{sync}} + \lambda I$), and $V_{t_p+E_p-2,i} = \overline{V}_{t_p+E_p-2,i} + g(x_{t,i};\theta_0)g(x_{t,i};\theta_0)^\top / m$, and also used $\det \overline{V}_{t_p+E_p-2,i} \le \det V_{t_p+E_p-2,i}$. To understand step $(d)$, note that the term in step $(c)$: $((t_p + E_p - 2) - t_{\text{last}}) \log \frac{\det V_{t_p+E_p-2,i}}{\det V_{\text{last}}}$ is exactly the criterion we use to check whether to start a communication round in iteration $t = t_p + E_p - 2$ (line 11 of Algo. 1). Since $t = t_p + E_p - 2$ is not the last iteration in this epoch (i.e., we did not start a communication round after checking this criterion in iteration $t = t_p + E_p - 2$), therefore, this criterion is not satisfied in iteration $t = t_p + E_p - 2$, i.e., $((t_p + E_p - 2) - t_{\text{last}}) \log \frac{\det V_{t_p+E_p-2,i}}{\det V_{\text{last}}} \le D$, which explains step $(d)$.

Next, we can plug equation 21 into equation 20 to obtain:

$$R^{[p]} = \sum_{i=1}^{N} \sum_{t=t_p}^{t_p+E_p-1} r_{t,i} \le \sum_{i=1}^{N} \left( 4 + 2\nu_{TKN}\sqrt{\kappa_0 \lambda}\sqrt{2D} \right) = \left( 4 + 2\nu_{TKN}\sqrt{2\kappa_0 \lambda D} \right) N, \quad (22)$$

which gives an upper bound on the total regret from *any* bad epoch. Now recall that as we have discussed in App. C.2, there are no more than $\overline{R}$ bad epochs (with probability of at least $1 - \delta_1$). Therefore, the total regret of *all* bad epochs can be upper-bounded by:

$$
\begin{aligned}
R_T^{\text{bad}} &\le \overline{R}\left( 4 + 2\nu_{TKN}\sqrt{2\kappa_0 \lambda D} \right) N \\
&\le \left( \widetilde{d}\log(1 + TKN/\lambda) + 1 \right)\left( 4 + 2\nu_{TKN}\sqrt{2\kappa_0 \lambda D} \right) N \\
&= \widetilde{O}\left( \widetilde{d}\sqrt{\widetilde{d}}\sqrt{D}N \right) \\
&= \widetilde{O}\left( (\widetilde{d})^{3/2}\sqrt{D}N \right).
\end{aligned}
\quad (23)
$$

In the second last equality, we have used $\nu_{TKN} = \widetilde{O}(\sqrt{\widetilde{d}})$. By choosing $D = \mathcal{O}(\frac{T}{N\widetilde{d}})$ (line 1 of Algo. 1), we can further express the above upper bound on the total regrets from all bad epochs as:

$$
\begin{aligned}
R_T^{\text{bad}} &= \mathcal{O}\left( \sqrt{\frac{T}{N\widetilde{d}}}(\widetilde{d})^{3/2}N \right) \\
&= \mathcal{O}\left( \widetilde{d}\sqrt{TN} \right).
\end{aligned}
\quad (24)
$$

### C.6   FINAL REGRET UPPER BOUND

Here we derive an upper bound on the total cumulative regret by adding up the regrets resulting from all good epochs (App. C.4) and all bad epochs (App. C.5):

$$
\begin{aligned}
R_T &= R_T^{\text{good}} + R_T^{\text{bad}} \\
&= \widetilde{O}\left( \widetilde{d}\sqrt{TN} + \widetilde{d}_{\max}N\sqrt{T} + TN\varepsilon_{\text{linear}}(m,T) + \widetilde{d}\sqrt{TN} \right) \\
&= \widetilde{O}\left( \widetilde{d}\sqrt{TN} + \widetilde{d}_{\max}N\sqrt{T} + TN\varepsilon_{\text{linear}}(m,T) \right).
\end{aligned}
\quad (25)
$$

This regret upper bound holds with probability of at least $1 - \delta_1 - \delta_2 - \delta_3 - \delta_4 - \delta_5 - \delta_6$. We let $\delta_3 = \delta_4 = \delta/3$, which leads to the expressions of $\nu_{TKN}$ and $\nu_{TK}$ given in the main paper (Sec. 3). We let $\delta_1 = \delta_2 = \delta_5 = \delta_6 = \delta/12$, and this will only introduce an additional factor of $\log 12$ in the first three conditions on $m$ in App. C.1 which can be absorbed by the constant $C$.

Next, the last term from the upper bound in equation 25 can be further written as:

$$
\begin{aligned}
TN\varepsilon_{\text{linear}}(m,T) &= TN\left( \varepsilon_{\text{linear},1}(m,T) + \varepsilon_{\text{linear},2}(m,T) + \varepsilon_{\eta,J} \right) \\
&= TNC_1 T^{2/3}m^{-1/6}\lambda^{-2/3}L^3\sqrt{\log m} + TNC_3 m^{-1/6}\sqrt{\log m}L^4 T^{5/3}\lambda^{-5/3}(1 + \sqrt{T/\lambda}) \\
&\quad + TNC_2(1 - \eta m\lambda)^J \sqrt{TL/\lambda}.
\end{aligned}
$$
$$(26)$$

It can be easily verified that as long as $m(\log m)^{-3} \geq 3^6 C_1^6 T^{10} N^6 \lambda^{-4} L^{18}$ and $m(\log m)^{-3} \geq 3^6 C_3^6 T^{16} N^6 L^{24} \lambda^{-10} (1 + \sqrt{T/\lambda})^6$ (which are ensured by conditions 5 and 6 on $m$ in App. C.1), then the first and second terms in equation 26 can both be upper-bounded by $1/3$. Moreover, if the conditions on $\eta$ and $J$ presented in App. C.1 are satisfied, i.e., if we choose the learning rate as $\eta = C_4 (m\lambda + mTL)^{-1}$ in which $C_4 > 0$ is an absolute constant such that $C_4 \leq 1 + TL$, and choose $J = \frac{1}{C_4} \left(1 + \frac{TL}{\lambda}\right) \log \left(\frac{1}{3C_2 N} \sqrt{\frac{\lambda}{T^3 L}}\right) = \widetilde{O}\left(TL/(\lambda C_4)\right)$, then the third term in equation 26 can also be upper-bounded by $1/3$.

As a result, the last term from the upper bound in equation 25 can be upper-bounded by 1, and hence the regret upper bound becomes:

$$R_T = \widetilde{O}\Big(\widetilde{d}\sqrt{TN} + \widetilde{d}_{\max} N \sqrt{T}\Big). \tag{27}$$

**Worst-Case Regret Upper Bound in Terms of the Maximum Information Gain $\gamma$.** Next, we perform some further analysis of the final regret upper bound derived above, which allows us to inspect the order of growth of our regret upper bound in the worst-case scenario (i.e., without assuming that the effective dimensions are upper-bounded by constants). We have defined in Sec. 2 that $\widetilde{d} \leq 2\gamma_{TKN}/\log(1 + TKN/\lambda)$, $\widetilde{d}_i \leq 2\gamma_{TK}/\log(1 + TK/\lambda)$, $\forall i \in [N]$ and $\widetilde{d}_{\max} = \max_{i \in [N]} \widetilde{d}$. As a result, in our derivations in equation 19 and equation 23, we can replace $\widetilde{d} \log(1 + TKN/\lambda)$ by $2\gamma_{TKN}$ and replace $\widetilde{d}_j \log(1 + TK/\lambda)$ by $2\gamma_{TK}$, after which the regret upper bound becomes

$$R_T = \widetilde{O}\Big(\gamma_{TKN}\sqrt{TN} + \gamma_{TK} N \sqrt{T}\Big). \tag{28}$$

The growth rate of the maximum information gain of NTK has been characterized by previous works: $\gamma_T = \widetilde{\mathcal{O}}(T^{\frac{d-1}{d}})$ (Kassraie & Krause, 2022; Vakili et al., 2021). This implies that our regret upper bound can be further expressed as

$$R_T = \widetilde{O}\left(K^{\frac{(d-1)}{d}}(TN)^{\frac{3d-2}{2d}} + K^{\frac{(d-1)}{d}} T^{\frac{3d-2}{2d}} N\right) = \widetilde{O}\left(K^{\frac{(d-1)}{d}} T^{\frac{3d-2}{2d}} N\right).$$

## C.7 Regret Upper Bound for FN-UCB (Less Comm.)

Here we explain how the proof above can be modified to derive a regret upper bound FN-UCB (Less Comm.). To begin with, note that in terms of the regret analysis, the only difference between FN-UCB (Less Comm.) and FN-UCB is that $\mathrm{UCB}_{t,i}^b$ of every agent $i$ is now modified to be: $\mathrm{UCB}_{t,i}^b(x) = f(x; \theta_{\mathrm{sync,NN}}) + \nu_{TK}\sqrt{\lambda}\big\|g(x; \theta_0)/\sqrt{m}\big\|_{V_{\mathrm{sync,NN}}^{-1}}$, in which the matrix $V_{\mathrm{sync,NN}}^{-1}$ is obtained by: $V_{\mathrm{sync,NN}}^{-1} = \frac{1}{N} \sum_{i=1}^N (V_{t,i}^{\mathrm{local}})^{-1}$. Note that every time the matrix $V_{\mathrm{sync,NN}}^{-1}$ is calculated, we have that $t = t_p - 1$.

**Firstly**, we prove that the modified $\mathrm{UCB}_{t,i}^b$ is also a valid high-probability upper bound on the reward function $f$. To achieve this, all we need to do is to add a few steps to equation 11 in **Step 3** of the

proof of. Specifically, we can further analyze equation 11 by:

$$
\begin{aligned}
&|f(x;\theta_{\text{sync,NN}}) - h(x)| \\
&\leq \frac{1}{N}\sum_{i=1}^{N}\nu_{TK}\sqrt{\lambda}\big\|g(x;\theta_0)/\sqrt{m}\big\|_{(V_i^{\text{local}})^{-1}} + \varepsilon_{\text{linear}}(m,T) \\
&= \nu_{TK}\frac{1}{N}\sum_{i=1}^{N}\sqrt{\lambda g(x;\theta_0)^\top (V_i^{\text{local}})^{-1}g(x;\theta_0)/m} + \varepsilon_{\text{linear}}(m,T) \\
&\leq \nu_{TK}\sqrt{\frac{1}{N}\sum_{i=1}^{N}\lambda g(x;\theta_0)^\top (V_i^{\text{local}})^{-1}g(x;\theta_0)/m} + \varepsilon_{\text{linear}}(m,T) \\
&= \nu_{TK}\sqrt{\lambda g(x;\theta_0)^\top \left(\frac{1}{N}\sum_{i=1}^{N}(V_i^{\text{local}})^{-1}\right)g(x;\theta_0)/m} + \varepsilon_{\text{linear}}(m,T) \\
&= \nu_{TK}\sqrt{\lambda g(x;\theta_0)^\top \left(V_{\text{sync,NN}}^{-1}\right)g(x;\theta_0)/m} + \varepsilon_{\text{linear}}(m,T) \\
&= \nu_{TK}\sqrt{\lambda}\big\|g(x;\theta_0)/\sqrt{m}\big\|_{V_{\text{sync,NN}}^{-1}} + \varepsilon_{\text{linear}}(m,T).
\end{aligned}
\tag{29}
$$

The first inequality directly follows from equation 11, and the second inequality results from the concavity of the square root function. In the second last equality, we have plugged in the definition of $V_{\text{sync,NN}}^{-1} = \frac{1}{N}\sum_{i=1}^{N}(V_{t,i}^{\text{local}})^{-1}$. As a result, Lemma 4 which guarantees the validity of $\text{UCB}_{t,i}^{b}$ can be modified to be:

$$
|h(x) - f(x;\theta_{\text{sync,NN}})| \leq \nu_{TK}\sqrt{\lambda}\big\|g(x;\theta_0)/\sqrt{m}\big\|_{V_{\text{sync,NN}}^{-1}} + \varepsilon_{\text{linear}}(m,T), \forall x \in \mathcal{X}_{t,i}.
\tag{30}
$$

**Secondly**, we will need the following auxiliary inequality for agent $i$ and iteration $t$ in a good epoch $p \in \mathcal{E}^{\text{good}}$:

$$
\begin{aligned}
\sqrt{\lambda}\big\|g(x_{t,i};\theta_0)/\sqrt{m}\big\|_{V_{\text{sync,NN}}^{-1}} &= \sqrt{\lambda g(x_{t,i};\theta_0)^\top V_{\text{sync,NN}}^{-1}g(x_{t,i};\theta_0)/m} \\
&= \sqrt{\lambda g(x_{t,i};\theta_0)^\top \left(\frac{1}{N}\sum_{j=1}^{N}(V_j^{\text{local}})^{-1}\right)g(x_{t,i};\theta_0)/m} \\
&= \sqrt{\frac{1}{N}\sum_{j=1}^{N}\lambda g(x_{t,i};\theta_0)^\top (V_j^{\text{local}})^{-1}g(x_{t,i};\theta_0)/m} \\
&\leq \frac{1}{\sqrt{N}}\sum_{j=1}^{N}\sqrt{\lambda g(x_{t,i};\theta_0)^\top (V_j^{\text{local}})^{-1}g(x_{t,i};\theta_0)/m} \\
&\leq \frac{1}{\sqrt{N}}\sum_{j=1}^{N}\sqrt{\lambda}\big\|g(x_{t,i};\theta_0)/\sqrt{m}\big\|_{(V_j^{\text{local}})^{-1}}.
\end{aligned}
\tag{31}
$$

The first inequality is because $\sqrt{a+b} \leq \sqrt{a} + \sqrt{b}$.

**Thirdly**, we need to modify the proof of the regret upper bound for good epochs (App. C.4). Specifically, we can derive an upper bound on the instantaneous regret $r_{t,i} = h(x_{t,i}^*) - h(x_{t,i})$ for

agent $i$ and iteration $t$ in a good epoch $p \in \mathcal{E}^{\text{good}}$ (in a similar way to equation 13):

$$
\begin{aligned}
r_{t,i} &= h(x_{t,i}^*) - h(x_{t,i}) \\
&\leq \alpha\Big(2\nu_{TK}\sqrt{\lambda}\big\|g(x_{t,i};\theta_0)/\sqrt{m}\big\|_{V_{\text{sync,NN}}^{-1}} + \varepsilon_{\text{linear}}(m,T)\Big) + \\
&\quad (1-\alpha)\Big(2\nu_{TKN}\sqrt{\lambda}\big\|g(x_{t,i};\theta_0)/\sqrt{m}\big\|_{\overline{V}_{t,i}^{-1}}\Big) + \alpha\varepsilon_{\text{linear}}(m,T) \\
&\leq \alpha\Big(2\nu_{TK}\frac{1}{\sqrt{N}}\sum_{j=1}^N\sqrt{\lambda}\big\|g(x_{t,i};\theta_0)/\sqrt{m}\big\|_{(V_j^{\text{local}})^{-1}} + \varepsilon_{\text{linear}}(m,T)\Big) + \\
&\quad (1-\alpha)\Big(2\nu_{TKN}\sqrt{e\lambda}\big\|g(x_{t,i};\theta_0)/\sqrt{m}\big\|_{\widetilde{V}_{t,i}^{-1}}\Big) + \alpha\varepsilon_{\text{linear}}(m,T) \\
&= \alpha 2\nu_{TK}\frac{1}{\sqrt{N}}\sum_{j=1}^N\sqrt{\lambda}\big\|g(x_{t,i};\theta_0)/\sqrt{m}\big\|_{(V_j^{\text{local}})^{-1}} + \\
&\quad (1-\alpha)2\nu_{TKN}\sqrt{e\lambda}\big\|g(x_{t,i};\theta_0)/\sqrt{m}\big\|_{\widetilde{V}_{t,i}^{-1}} + 2\alpha\varepsilon_{\text{linear}}(m,T) \\
&\triangleq (1-\alpha)2\nu_{TKN}\sqrt{e}\widetilde{\sigma}_{t,i}(x_{t,i}) + \alpha 2\nu_{TK}\frac{1}{\sqrt{N}}\sum_{j=1}^N\widetilde{\sigma}_{t_p-1,j}^{\text{local}}(x_{t,i}) + 2\alpha\varepsilon_{\text{linear}}(m,T).
\end{aligned}
\tag{32}
$$

In the first inequality, we have made use of equation 30 which ensures the validity of the modified $\text{UCB}_{t,i}^b$ as a high probability upper bound on $h$. The second inequality follows from equation 31. In the last equality, we have defined $\widetilde{\sigma}_{t_p-1,j}^{\text{local}}(x_{t,i})$ in the same way as equation 13. The steps regarding the term involving $(1-\alpha)$ are the same as those from equation 13. As a result, the only change we have made to instantaneous regret upper bound from equation 13 is that in the second term, we have replaced $\frac{1}{N}$ by $\frac{1}{\sqrt{N}}$. Further propagating this change through the proof for the regret upper bound for all good epochs (App. C.4.4 and App. C.4.5), we have that:

$$
R_T^{\text{good}} = \sum_{i=1}^N\sum_{t\in\mathcal{T}^{\text{good}}}r_{t,i} = \widetilde{O}\left(\widetilde{d}\sqrt{TN} + \widetilde{d}_{\max}N^{3/2}\sqrt{T} + TN\varepsilon_{\text{linear}}(m,T)\right).
\tag{33}
$$

**Lastly**, also note that the regret upper bound for the bad epochs (i.e., the proof in App. C.5) remains unchanged. Therefore, the final regret upper bound for `FN-UCB (Less Comm.)` is

$$
\begin{aligned}
R_T &= R_T^{\text{good}} + R_T^{\text{bad}} \\
&= \widetilde{O}\Big(\widetilde{d}\sqrt{TN} + \widetilde{d}_{\max}N^{3/2}\sqrt{T} + TN\varepsilon_{\text{linear}}(m,T) + \widetilde{d}\sqrt{TN}\Big) \\
&= \widetilde{O}\Big(\widetilde{d}\sqrt{TN} + \widetilde{d}_{\max}N^{3/2}\sqrt{T} + TN\varepsilon_{\text{linear}}(m,T)\Big) \\
&= \widetilde{O}\Big(\widetilde{d}\sqrt{TN} + \widetilde{d}_{\max}N^{3/2}\sqrt{T}\Big).
\end{aligned}
\tag{34}
$$

# D  PROOF OF UPPER BOUND ON COMMUNICATION COMPLEXITY (THEOREM 2)

In this section, we derive an upper bound on the communication complexity (i.e., the total number of communication rounds) of our `FN-UCB` algorithm (including its variant `FN-UCB (Less Comm.)`). Define $\zeta \triangleq \sqrt{DT/R}$. An immediate implication is that there can be at most $\lceil T/\zeta\rceil$ epochs whose length is larger than $\zeta$. Next, we try to derive an upper bound on the number of epochs whose length is smaller than $\zeta$.

Note that if an epoch $p$ contains less than $\zeta$ iterations, then because of our criterion to start a communication round (line 10 of Algo. 1), we have that $\log\frac{\det V_p}{\det V_{p-1}} > \frac{D}{\zeta}$. Also recall that equation equation 4 (Appendix C.2) tells us that:

$$
\sum_{p=0}^{P-1}\log\frac{\det V_{p+1}}{\det V_p} \leq R' \leq \overline{R},
\tag{35}
$$

with probability of at least $1 - \delta_1 \geq 1 - \delta$. Therefore, there can be at most $\lceil \frac{\overline{R}}{D/\zeta} \rceil = \lceil \frac{\overline{R}\zeta}{D} \rceil$ such epochs whose length is smaller than $\zeta$. As a result, the total number of epochs can be upper-bounded by:

$$\lceil T/\zeta \rceil + \lceil \frac{\overline{R}\zeta}{D} \rceil = \mathcal{O}\Big(\sqrt{\frac{T\overline{R}}{D}}\Big). \tag{36}$$

Recall that $\overline{R} = \widetilde{\mathcal{O}}(\widetilde{d})$ (App. C.2). Therefore, with probability of at least $1 - \delta_1 \geq 1 - \delta$, the total number of epochs can be upper-bounded by $\widetilde{\mathcal{O}}(\sqrt{\frac{T\widetilde{d}}{D}})$.

Since we have chosen $D = \widetilde{\mathcal{O}}(\frac{T}{N\widetilde{d}})$ (line 1 of Algo. 1), therefore, the total number of epochs can be upper-bounded by $\widetilde{\mathcal{O}}(\sqrt{\frac{T\widetilde{d}}{\frac{T}{N\widetilde{d}}}}) = \widetilde{\mathcal{O}}(\widetilde{d}\sqrt{N})$. Now we can further make use of the relationship between $\widetilde{d}$ and $\gamma_{TKN}$: $\widetilde{d} \leq 2\gamma_{TKN}/\log(1 + TKN/\lambda)$, which allows us to show that the worst-case communication complexity is upper-bounded by: $\widetilde{\mathcal{O}}(\widetilde{d}\sqrt{N}) = \widetilde{\mathcal{O}}\left(\gamma_{TKN}\sqrt{N}\right) = \widetilde{\mathcal{O}}\left((TKN)^{\frac{d-1}{d}}\sqrt{N}\right) = \widetilde{\mathcal{O}}(T^{\frac{d-1}{d}}K^{\frac{d-1}{d}}N^{\frac{3d-2}{2d}})$, which is still sub-linear in $T$ even in the worst case.

The proof here, and hence Theorem 2, makes use of Lemma 1. Therefore, we only need condition 1 on $m$ listed in App. C.1 to hold, and do not require any condition on $\eta$ and $J$.

# E  MORE EXPERIMENTAL DETAILS

Our code can be found at: `https://github.com/daizhongxiang/Federated-Neural-Bandits`.

Some of the experimental details (e.g., the number of layers and the width $m$ of the NN used in every experiment) are already described in the main text (Sec. 5). Following the works of Zhang et al. (2021); Zhou et al. (2020), when training the NN (line 14 of Algo. 1) for agent $i$, we use the NN parameters resulting from the last gradient descent training of agent $i$ (instead of $\theta_0$) as the initial parameters, in order to accelerate the training procedure. Every time we train an NN, we use stochastic gradient descent to train the NN for 30 iterations with a learning rate of 0.01. To save computational cost, we stop training the NNs after 2000 iterations, i.e., after 2000 iterations, all NN parameters are no longer updated. Also to reduce the computational cost, when checking the criterion in line 11 of Algo. 1, we diagonalize (i.e., only keep the diagonal elements of) the two matrices for which we need to calculate the determinant. Our experiments are run on a server with 96 CPUs, an NVIDIA A100 GPU with a memory of 40GB, a RAM of 256GB, running the Ubuntu system.

The `shuttle` dataset is publicly available at `https://archive.ics.uci.edu/ml/datasets/Statlog+(Shuttle)` and contains no personally identifiable information or offensive content. It includes 58000 instances, has an input dimension of $d = 9$ and contains $K = 7$ classes/arms. As a result, according to the way in which the contexts are constructed (Sec. 5.2), every context feature vector has a dimension of $9 \times 7 = 63$. The `magic telescope` dataset is publicly available at `https://archive.ics.uci.edu/ml/datasets/magic+gamma+telescope` and contains no personally identifiable information or offensive content. The dataset contains 19020 instances, has an input dimension of $d = 10$ and $K = 2$ classes/arms. As a result, every context feature vector has a dimension of $10 \times 2 = 20$.

When comparing with Linear-UCB, Linear TS, Kernelized UCB and Kernelized TS, we follow the work of Zhang et al. (2021) to set $\lambda = 1$ and perform a grid search within $\nu \in \{1, 0.1, 0.01\}$. The results showing comparisons with these algorithms, for both the synthetic experiments (Sec. 5.1) and real-world experiments (Sec. 5.2), are presented in Fig. 4. The figures show that both linear and kernelized contextual bandit algorithms are outperformed by neural contextual bandit algorithms, which is consistent with the observations from Zhang et al. (2021); Zhou et al. (2020).

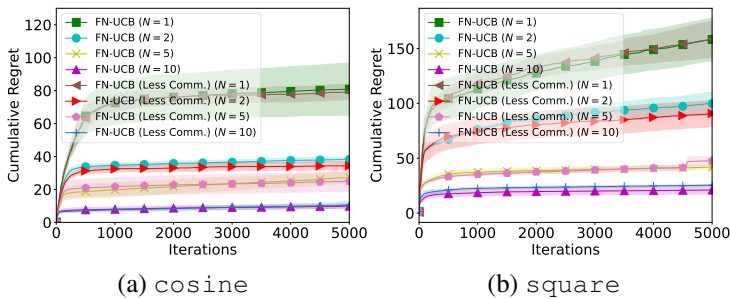

Figure 3: Cumulative regret of `FN-UCB` and `FN-UCB (Less Comm.)` for the `cosine` and `square` functions. Their performances are very similar for both functions.

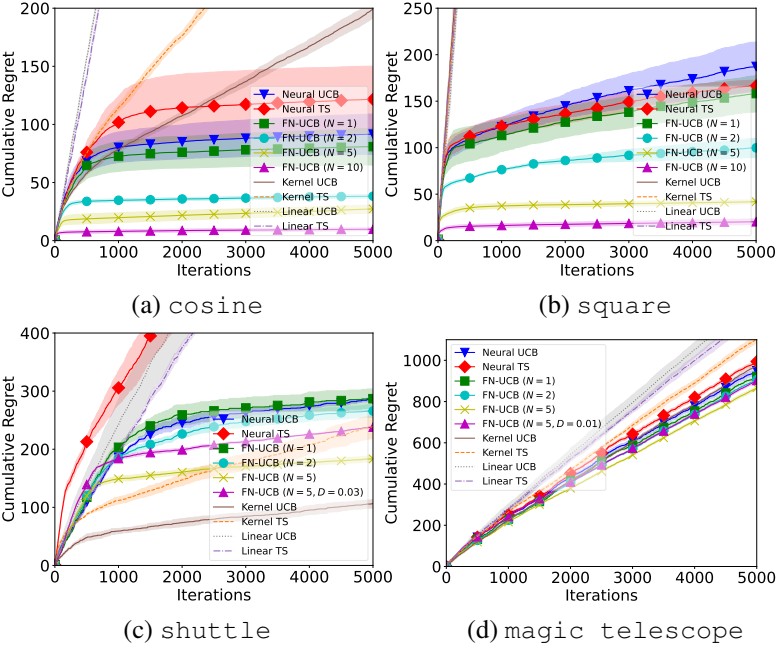

Figure 4: Cumulative regrets for the (a) `cosine`, (b) `square`, (c) `shuttle` (with diagonalization), and (d) `magic telescope` experiments, with additional comparisons with Linear UCB, Linear TS, Kernel UCB and Kernel TS.

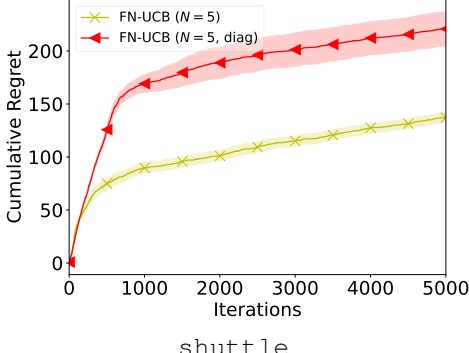

shuttle

Figure 5: Comparison between the performances without (yellow) and with (red) diagonalization, using $m = 20$ with the `shuttle` dataset. The results show that using an NN with the same width $m = 20$, diagonalization indeed deteriorates the performances.

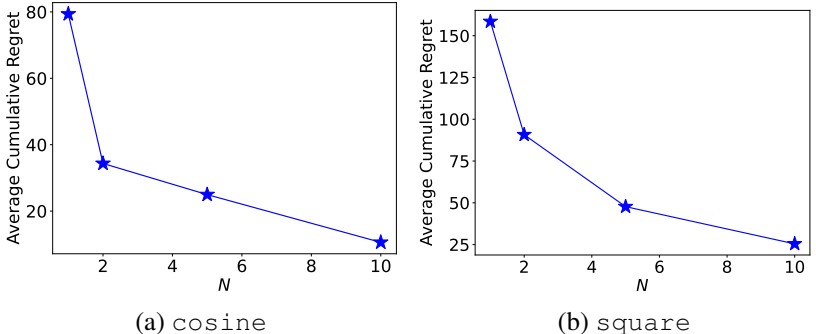

(a) `cosine`                    (b) `square`

Figure 6: The scaling of the final average cumulative regret after 5000 iterations (averaged across all $N$ agents) in terms of the number $N$ of agents, using the `cosine` and `square` experiments. The results correspond to Fig. 1 a and Fig. 1 b, respectively.

We have additionally evaluated the empirical impact of the technique of diagonalization of the matrices (Sec. 3.4), using the `shuttle` dataset and a fixed width of $m = 20$ for the NN. The results (Fig. 5) show that for the same width of the NN, the technique of diagonalization indeed results in worse performances. However, also note that diagonalization allows us to afford a larger value of $m$ in a computationally feasible way, which can lead to better performances than using a smaller $m$ without diagonalization. This is corroborated by our empirical results in Fig. 2a and 2c, because the regrets in Fig. 2c ($m$=50 , with diagonalization) are in general smaller than the regrets in Fig. 2a ($m$=20 , without diagonalization), and the computational cost of Fig. 2c (244.9 seconds) is smaller than that of Fig. 2a (361.8 seconds). Furthermore, using $m = 50$ without diagonalization would incur a significantly larger computational cost (3134.3 seconds). These results demonstrate the practical usefulness of diagonalization.

We have also visualized the empirical scaling of the final average cumulative regret (after 5000 iterations) in terms of the number $N$ of agents, using the `cosine` and `square` experiments. The results (Fig. 6) demonstrate that the average cumulative regret (averaged across all $N$ agents) is indeed decreasing as the number $N$ of agents increases.

# F    EXTENDED ANALYSIS FOR THE GENERAL ALGORITHM

Recall that it has been mentioned at the beginning of Sec. 4.1 that our main regret analysis (Theorem 1) has focused on a simpler version of our `FN-UCB` algorithm, in which we only choose the value of $\alpha$ using the method described in Sec. 3.3 in the first iteration of every epoch and set $\alpha = 0$ in the other iterations. Here, we show how our regret analysis can be extended to derive a regret upper bound

for the general `FN-UCB` algorithm, in which we choose $\alpha$ using the method described in Sec. 3.3 in every iteration, i.e., we do not set $\alpha = 0$ in any iteration. To achieve this, we need an additional assumption of an upper bound on the amount of new information collected by every agent $i$ in every epoch $p$. Specifically, we assume that

$$\frac{\det V^{\text{local}}_{t_p+E_p-2,i}}{\det V^{\text{local}}_{t_p-1,i}} \leq \overline{D}, \forall i \in [N], p \in [P] \tag{37}$$

for a constant $\overline{D} \geq 1$. This can in fact be viewed as an additional property of the sequence of contexts for each agent. Intuitively, if the contexts for each agent are received in such an order that similar contexts also arrive in similar iterations, then the constant $\overline{D}$ is likely to be small. This can be seen as a "stationarity" property of the sequence of contexts, which is reasonable in many practical scenarios. For example, in a healthcare application, the patients arriving within the same time period are likely to have similar characteristics due to factors such as the local transmission of a seasonal flu. In addition, another scenario where $\overline{D}$ is likely to be small is when every agent has some previously observed offline contexts before running our algorithm. If these offline contexts have a good coverage of the space of contexts, then conditioned on these offline contexts, the newly collected information by every agent in every epoch is highly likely to be small.

With this additional assumption, the most important step in the proof that we need to modify is the proof in Appendix C.4.4, in which we proved an upper bound on the sum of the second term of equation 13. To begin with, $\forall t = t_p, \ldots, t_p + E_p - 1$, we have that

$$\begin{aligned}
\widetilde{\sigma}^{\text{local}}_{t_p-1,j}(x_{t,i}) &\overset{(a)}{=} \sqrt{\lambda}\big\|g(x_{t,i};\theta_0)/\sqrt{m}\big\|_{(V^{\text{local}}_{t_p-1,j})^{-1}} \\
&= \sqrt{\lambda g(x_{t,i};\theta_0)^\top (V^{\text{local}}_{t_p-1,j})^{-1} g(x_{t,i};\theta_0)/m} \\
&\overset{(b)}{\leq} \sqrt{\lambda g(x_{t,i};\theta_0)^\top (V^{\text{local}}_{t-1,j})^{-1} g(x_{t,i};\theta_0)/m \frac{\det V^{\text{local}}_{t-1,j}}{\det V^{\text{local}}_{t_p-1,j}}} \\
&\overset{(c)}{\leq} \sqrt{\lambda g(x_{t,i};\theta_0)^\top (V^{\text{local}}_{t-1,j})^{-1} g(x_{t,i};\theta_0)/m \frac{\det V^{\text{local}}_{t_p+E_p-1-1,j}}{\det V^{\text{local}}_{t_p-1,j}}} \\
&\overset{(d)}{\leq} \sqrt{\lambda g(x_{t,i};\theta_0)^\top (V^{\text{local}}_{t-1,j})^{-1} g(x_{t,i};\theta_0)/m\overline{D}} \\
&\overset{(e)}{=} \sqrt{\overline{D}}\widetilde{\sigma}^{\text{local}}_{t-1,j}(x_{t,i}).
\end{aligned} \tag{38}$$

Step $(a)$ has made use of the definition of $\widetilde{\sigma}^{\text{local}}_{t_p-1,j}(x_{t,i})$ (see the paragraph below equation 13), step $(b)$ results from Lemma 12 of Abbasi-Yadkori et al. (2011), step $(c)$ follows because $V^{\text{local}}_{t_p+E_p-1-1,j}$ contains more information than $V^{\text{local}}_{t-1,j}$ $\forall t = t_p, \ldots, t_p + E_p - 1$, step $(d)$ follows from equation 37, and step $(e)$ has again made use of the definition of $\widetilde{\sigma}^{\text{local}}_{t_p-1,j}(x_{t,i})$.

Using equation 38, we can modify the proof in equation 16 (Appendix C.4.4):

$$\sum_{i=1}^{N} \sum_{p \in \mathcal{E}^{\text{good}}} \sum_{t \in \mathcal{T}^{(p)}} \alpha 2\nu_{TK} \frac{1}{N} \sum_{j=1}^{N} \widetilde{\sigma}_{t_p-1,j}^{\text{local}}(x_{t,i}) \le 2\nu_{TK} \frac{1}{N} \sum_{i=1}^{N} \sum_{p \in [P]} \sum_{t \in \mathcal{T}^{(p)}} \alpha \sum_{j=1}^{N} \widetilde{\sigma}_{t_p-1,j}^{\text{local}}(x_{t,i})$$

$$\le 2\nu_{TK} \frac{1}{N} \sum_{i=1}^{N} \sum_{j=1}^{N} \sum_{p \in [P]} \sum_{t=t_p}^{t_p+E_p-1} \alpha \widetilde{\sigma}_{t_p-1,j}^{\text{local}}(x_{t,i})$$

$$\overset{(a)}{\le} 2\nu_{TK} \frac{1}{N} \sum_{i=1}^{N} \sum_{j=1}^{N} \sum_{p \in [P]} \sum_{t=t_p}^{t_p+E_p-1} \widetilde{\sigma}_{t_p-1,j}^{\text{local}}(x_{t,j}) \qquad (39)$$

$$\overset{(b)}{\le} 2\nu_{TK} \frac{1}{N} \sum_{i=1}^{N} \sum_{j=1}^{N} \sum_{p \in [P]} \sum_{t=t_p}^{t_p+E_p-1} \sqrt{\overline{\overline{D}}} \widetilde{\sigma}_{t-1,j}^{\text{local}}(x_{t,j})$$

$$= \sqrt{\overline{\overline{D}}} 2\nu_{TK} \frac{1}{N} \sum_{i=1}^{N} \sum_{j=1}^{N} \sum_{t=1}^{T} \widetilde{\sigma}_{t-1,j}^{\text{local}}(x_{t,j}).$$

Step $(a)$ follows from the same reasoning as step $(c)$ of equation 16, step $(b)$ has made use of equation 38, and all other steps follow the same corresponding steps of equation 16.

As a result, by comparing the modified equation 39 with the original equation 16, the only modification to the result in equation 16 is the additional multiplicative term of $\sqrt{\overline{\overline{D}}}$. Therefore, after propagating this modification to all the analysis in Appendix C.4.4, we have that a multiplicative term of $\sqrt{\overline{\overline{D}}}$ will also be introduced into equation 18. Subsequently, the upper bound on the total regrets from all good epochs (i.e., equation 19) will be correspondingly modified to be:

$$R_T^{\text{good}} = \widetilde{O}\left( \widetilde{d}\sqrt{TN} + \sqrt{\overline{\overline{D}}} \widetilde{d}_{\max} N\sqrt{T} + TN\varepsilon_{\text{linear}}(m, T) \right). \qquad (40)$$

Next, we also need to modify the proof of the upper bound on the total regrets from all *bad epochs* (Appendix C.5). Following the roadmap of Appendix C.5, we start by upper-bounding the total regrets from a particular bad epoch $p$:

$$R^{[p]} = \sum_{i=1}^{N} \sum_{t=t_p}^{t_p+E_p-1} r_{t,i} = \sum_{i=1}^{N} \sum_{t=t_p}^{t_p+E_p-1} [\alpha h(x_{t,i}^*) + (1-\alpha)h(x_{t,i}^*) - h(x_{t,i})]$$

$$\overset{(a)}{\le} \sum_{i=1}^{N} \sum_{t=t_p}^{t_p+E_p-1} \left[ \alpha \text{UCB}_{t,i}^b(x_{t,i}^*) + \alpha\varepsilon_{\text{linear}}(m, T) + (1-\alpha)\text{UCB}_{t,i}^a(x_{t,i}^*) - h(x_{t,i}) \right]$$

$$\overset{(b)}{\le} \sum_{i=1}^{N} \sum_{t=t_p}^{t_p+E_p-1} \left[ \alpha \text{UCB}_{t,i}^b(x_{t,i}) + \alpha\varepsilon_{\text{linear}}(m, T) + (1-\alpha)\text{UCB}_{t,i}^a(x_{t,i}) - h(x_{t,i}) \right]$$

$$= \sum_{i=1}^{N} \sum_{t=t_p}^{t_p+E_p-1} \left[ \alpha(\text{UCB}_{t,i}^b(x_{t,i}) - h(x_{t,i})) + (1-\alpha)(\text{UCB}_{t,i}^a(x_{t,i}) - h(x_{t,i})) \right.$$

$$\left. + \alpha\varepsilon_{\text{linear}}(m, T) \right] \qquad (41)$$

$$\overset{(c)}{\le} \underbrace{\sum_{i=1}^{N} \left[ 4 + \sum_{t=t_p+1}^{t_p+E_p-2} \left( \text{UCB}_{t,i}^a(x_{t,i}) - h(x_{t,i}) \right) \right]}_{A}$$

$$+ \underbrace{\sum_{i=1}^{N} \sum_{t=t_p}^{t_p+E_p-1} \left[ \alpha(\text{UCB}_{t,i}^b(x_{t,i}) - h(x_{t,i})) + \alpha\varepsilon_{\text{linear}}(m, T) \right]}_{B}.$$

Step $(a)$ follows from Lemma 3 (i.e., the validity of $\text{UCB}_{t,i}^a$) and Lemma 4 (i.e., the validity of $\text{UCB}_{t,i}^b$). Step $(b)$ results from the way in which $x_{t,i}$ is selected (line 7 of Algo. 1): $x_{t,i} = \arg\max_{x \in \mathcal{X}_{t,i}} (1 - \alpha) \text{UCB}_{t,i}^a(x) + \alpha \text{UCB}_{t,i}^b(x)$. For step $(c)$, the term $A$ is obtained by upper-bounding the regrets of the first and last iteration within this epoch by 2 and using the fact that $\alpha \le 1$.

Next, we can separately analyze the terms $A$ and $B$ in equation 41. Firstly, note that the term $A$ is the same as step $(c)$ of equation 20, therefore, we can follow the same steps of analyses in App. C.5 (i.e., equation 20, equation 21, equation 22, equation 23 and equation 24) to show that after summing the term $A$ across all bad epochs, we get an upper bound of $\mathcal{O}\left(\widetilde{d}\sqrt{TN}\right)$. Secondly, for the term $B$, we can in fact follow similar steps of analysis in equation 13 to show that every term inside the square bracket of the term $B$ is upper-bounded by the last two terms in equation 13. That is,

$$\alpha(\text{UCB}_{t,i}^b(x_{t,i}) - h(x_{t,i})) + \alpha\varepsilon_{\text{linear}}(m,T) \le \alpha 2\nu_{TK} \frac{1}{N} \sum_{j=1}^{N} \widetilde{\sigma}_{t_p-1,j}^{\text{local}}(x_{t,i}) + 2\alpha\varepsilon_{\text{linear}}(m,T). \quad (42)$$

As a result, we can follow the same steps of analysis in App. C.4.4 (after making the modification using equation 39; note that the analysis in App. C.4.4 is applicable to both good and bad epochs) to show that after summing over all bad epochs, the term $B$ can be upper-bounded by $\widetilde{O}\left(\sqrt{\overline{D}}\widetilde{d}_{\max}N\sqrt{T} + TN\varepsilon_{\text{linear}}(m,T)\right)$. Next, combining the upper bounds on both $A$ and $B$ (after summing across all bad epochs), we have that for the general algorithm, the total regrets from all bad epochs can be upper bounded by

$$R_T^{\text{bad}} = \widetilde{O}\left(\widetilde{d}\sqrt{TN} + \sqrt{\overline{D}}\widetilde{d}_{\max}N\sqrt{T} + TN\varepsilon_{\text{linear}}(m,T)\right), \quad (43)$$

which is in fact the same as the upper bound on the total regrets from all good epochs which we have derived in equation 40.

Finally, following the same analysis in App. C.6, we can show that the final regret upper bound for the general algorithm, in which we do not set $\alpha = 0$ in any iteration, is

$$R_T = \widetilde{O}\left(\widetilde{d}\sqrt{TN} + \sqrt{\overline{D}}\widetilde{d}_{\max}N\sqrt{T}\right). \quad (44)$$

Note that compared to our regret upper bound from Theorem 1, the regret upper bound for the general algorithm (i.e., when we choose the value of $\alpha$ using the method from Sec. 3.3 in every iteration) only includes an additional multiplicative term of $\sqrt{\overline{D}}$ in the second term. Of note, when communication indeed occurs after each iteration (i.e., $E_p = 1$ for every epoch $p$), we have that $\overline{D} = 1$ because $\frac{\det V_{t_p+E_p-2,i}^{\text{local}}}{\det V_{t_p-1,i}^{\text{local}}} = 1$ (equation 37). In this case, the version of our algorithm analyzed in Theorem 1 becomes the same as our general algorithm (Sec. 4.1), and interestingly, the regret upper bound of our general algorithm (equation 44) also becomes the same as Theorem 1 because $\overline{D} = 1$.

