# OpenReview forum: "Federated Neural Bandits"
_ICLR.cc/2023/Conference — ICLR 2023 poster_

### Official Review · Reviewer_H38q · 2022-10-23

**Confidence:** 3
**Correctness:** 4
**Technical Novelty And Significance:** 2
**Empirical Novelty And Significance:** 2
**Recommendation:** 6

**Clarity, Quality, Novelty And Reproducibility:**

Clarity: It is easy to follow.
Quality and Novelty: OK, but not strong.
Reproducibility: The proof is detailed and the code is provided.



**Strength And Weaknesses:**

Strength:

1. The paper is well written and is very easy to follow.

2. The considered federated neural bandit problem is interesting.

3. Extensive experiments demonstrate the practical usefulness of the proposed algorithm.


Weaknesses:

1. Technical novelty: My major concern is its technical contribution beyond existing literature. The proposed federated neural bandit is a natural combination of neural bandit (Zhou et al., 2020; Zhang et al., 2021; Kassraie & Krause, 2022) and federated learning (Wang et al., 2020). Because of this nature, the algorithmic development and theoretical analysis were largely based on these two areas. In particular, a large part of the proof is to verify conditions used in Zhou et al. (2020) and Zhang et al. (2021). Based on my personal opinion, the technical novelty of this paper beyond these references is not strong.


2. Limitation in the current theory: There is a major gap in the theoretical results. The derived main theorems is not for the proposed FN-UCB, but for a simpler version where they only choose the weight $\alpha$ using the proposed method in the first iteration after every communication round and set $\alpha = 0$ in all other iterations. This simpler version requires communication to occur after each iteration, which clearly contradicts with the communication complexity shown in Theorem 2. Since the balance of $\alpha$ is a key novelty of the considered problem, it would be more convincing to derive the theoretical results for the proposed FN-UCB directly.


**Summary Of The Paper:**

This paper develops a federated neural contextual bandit algorithm FN-UCB which extends existing neural contextual bandits to the federated setting. The key idea of FN-UCB is that it adopts a weighted combination of two UCBs, where the first UCB allows every agent to additionally use the observations from the other agents to accelerate exploration, while the second UCB uses an NN for reward prediction. In theory, the regret bound of the proposed method is proved and its superior advantages are compared in extensive experiments.


**Summary Of The Review:**

The considered federated neural bandit is a natural combination of neural bandit and federated learning. The idea is very natural and the technical novelty is OK but not strong. Moreover, the current theoretical results only hold for a simpler version of the proposed FN-UCB (require communication in each iteration).

In the rebuttal stage, I would like to see more justifications on the technical novelty beyond existing neural bandit and federated learning, as well as an improved analysis for the general FN-UCB algorithm.

---

### Official Review · Reviewer_f21Q · 2022-10-30

**Confidence:** 3
**Correctness:** 4
**Technical Novelty And Significance:** 3
**Empirical Novelty And Significance:** 4
**Recommendation:** 8

**Clarity, Quality, Novelty And Reproducibility:**

Since this is the first work to apply NNs in a federated framework, the work appears novel. The proposed algorithm and supporting evidence seem solid to me, and the paper is well-written.

**Details Of Ethics Concerns:**

I do not have any concerns.

**Strength And Weaknesses:**

Strength
* This author(s) are the first to use neural netorks (NNs) in the setting of federated contextual bandits. Given that NNs have high representational capabilities, this opens up the possibility of more extensive real-world applications.
* The proposed algorithm, FN-UCB, has superior empirical performance that has been confirmed through extensive experiments in addition to theoretical assurance.

Weaknesses
* Although the author(s) have provided empirical experiments to demonstrate the proposed algorithms with multiple agents outperforms the baselines with single agent in practice, how efficient the suggested algorithm is is still unknown. It might be better if the author(s) could provide the lower bound.

**Summary Of The Paper:**

This research examines contextual bandits in a federated learning environment. This is the first study applying federated learning to neural contextual bandits. The author(s) proposed the FN-UCB algorithm, which is shown to suffer from sub-linear regret and communication rounds. Both synthetic and real-world experiments confirm that the newly proposed algorithm performs competitively against the baselines.

**Summary Of The Review:**

This author(s) are the first using neural netorks (NNs) in the setting of federated contextual bandits. Although the proposed algorithm may not be optimal, due to the NNs' powerful representational capabilities, broader real-world applications may be possible. Hence I suggest accept.

---

### Official Review · Reviewer_D1SJ · 2022-11-01

**Confidence:** 3
**Correctness:** 3
**Technical Novelty And Significance:** 3
**Empirical Novelty And Significance:** 2
**Recommendation:** 6

**Clarity, Quality, Novelty And Reproducibility:**

Overall the paper is clear and well-structured. The algorithm is easy to understand and properly analysed.

**Strength And Weaknesses:**

Overall the paper is clear and well-structured. The algorithm is easy to understand and properly analysed.

My only concerns are the experimental results provided by the authors, which somehow are not feeding each one of the analysed algorithms with the same number of samples at each round.

Details:
I would like you to discuss the results you obtained w.r.t. the federated bandit setting and the neural bandit ones. I think this would highlight how the elements of the regret and communication complexity change by the fact that you are using the combination of these two settings.

For instance, I would like to have the expression of the bound on the regret when you are considering only local information and comparing them with the ones you have in Theorem 2.

With N=1 agents does the regret of Theorem 2 become the one of Neural Bandits? Overall, you should discuss more on the difference between Neural Bandits and your algorithm with N=1.

In the first experiment (5.1) I think that the comparison is not fair. In this case, you are receiving two samples, while standard algorithms are receiving only one realization of the reward. However, similar comments are also holding for the second experiment.




Minor:
Please move the legend outside the figures, otherwise, they are hard to read.

**Summary Of The Paper:**

This work is analysing a new setting in which a set of multiple Neural Bandits are collaborating to minimize the regret for an online sequential decision problem. The authors provided an algorithm combining the feature of federated learning and neural bandits, with sublinear regret and communication cost. Finally, the authors demonstrate the capabilities of the developed methods with an experimental campaign.

**Summary Of The Review:**

The paper is sound and interesting, but some concerns about the experimental part should be addressed by the authors.

---

### Official Review · Reviewer_oNap · 2022-11-04

**Confidence:** 3
**Correctness:** 4
**Technical Novelty And Significance:** 3
**Empirical Novelty And Significance:** 4
**Recommendation:** 6

**Clarity, Quality, Novelty And Reproducibility:**

The paper is mostly clear with proof sketches for the main results. The problem setting incremental in the sense it is a natural extension of existing works but is sufficiently challenging and interesting to study.  Also, the main algorithms are shown in the paper and should be easily replicated.

**Strength And Weaknesses:**

Pros:
     (A) The proposed algorithm is a combination of two UCBs one of which is focused on exploration based on the neural tangent features. This has a higher weight
during the initial iterations. The second UCB kicks in once the neural networks are sufficiently trained and this corresponds to the exploitation phase. There is a
 nice explanation between these two phases. Combined with the regret guarantees over the iterations and communication rounds, they provide a good understanding and
 solution of the neural contextual bandit problem.
     (B) Extensive set of experiments where they switch off one of the UCBs gives us a good understanding of the individual components (ablation study). Also, the
performance based on setting of D as well the number of agents is explored.

Cons:

   (i) It was not fully clear on how alpha was set in the experiments? Section 3.3 describes the intuition as well as the mathematical formula for it but the experiments talk about a linear function. Any insight into this would be helpful. Also, the regret is analyzed for the setting where alpha=0 after first iteration. However, the experiments use non-zero alpha.
   (ii) Are there experiments which show the scaling with N or dtilde? It would be interesting to know how tight these bounds bound are and could serve as a proxy for lower-bounds for the setting.


**Summary Of The Paper:**

The authors propose a novel federated neural contextual bandit algorithm to tackle the richer structure of rewards that conventional linear/kernel approaches may not handle. Regret bounds for the proposed algorithms are derived and experimental results show the efficacy of the proposed algorithms on both synthetic and real-world experiments.


**Summary Of The Review:**

Overall, an interesting extension on the neural contextual bandit setting in the federated context. Some questions remain and I hope to discuss with the authors to clarify my understanding.

---

### Official Review · Reviewer_cGE9 · 2022-11-06

**Confidence:** 4
**Correctness:** 3
**Technical Novelty And Significance:** 3
**Empirical Novelty And Significance:** 3
**Recommendation:** 6

**Clarity, Quality, Novelty And Reproducibility:**

- [+] This pape is well-written and easy to follow. I did not find any reproducibility issues here.
- [-] The novelty of this paper is somehow limited, especially given the neural UCB and neural TS paper and the bunch of federated bandits paper.

**Strength And Weaknesses:**

### Strength

- This paper is well-written and easy to follow. The experiment is adequate and the proof is mathematically correct

### Weakness

- It seems that the major contribution made in this paper is combining the neural UCB and neural TS [1, 2] with federated bandits. The author might want to highlight their contributions and difficulties in this combination
- The claim on communication cost is somehow ambiguous. The authors suggest that by using the *Less Comm.* version of the algorithm, the communication cost could reduce to $O(p)$. However, the performance impact by 1) using the averaged covariance matrix $V$ instead of $V_i$ and 2) using the diagonal elements to approximate the matrix $V$, is not well studied either from a theoretical perceptive or empirical perceptive.

[1] Zhou, Dongruo, Lihong Li, and Quanquan Gu. "Neural contextual bandits with ucb-based exploration." International Conference on Machine Learning. PMLR, 2020.
[2] ZHANG, Weitong, et al. "Neural Thompson Sampling." International Conference on Learning Representations. 2020.

**Summary Of The Paper:**

This paper studies the federated bandits with neural networks under the NTK regime. Theoretical regret analysis and experimental results are provided to show that the proposed algorithm could fully leverage the neural networks in the federated bandit setting.

**Summary Of The Review:**

This paper is overall interesting and well-written. My major concern is the technical novelty of this paper as discussed above. Given this, I am on the borderline of rejection.


***

The author's response addressed some of my questions. Thus I raise my score accordingly

---

### Decision · Program_Chairs · 2023-01-20

**Decision:**

Accept: poster

**Justification For Why Not Higher Score:**

Some reviewers are still concerned that theoretical novelty is limited. The proof mainly combines existing analysis technical of federated bandits and NTK based neural bandits. No reviewer is willing to champion the paper.

**Justification For Why Not Lower Score:**

Reviewers are unanimously positive about the paper. Theoretical results are solid and experimental results are convincing. The proposed method and analysis are timely and could be interesting to the community.


**Metareview: Summary, Strengths And Weaknesses:**

The authors proposed the first federated neural bandit algorithm. The key idea of the proposed FN-UCB is combining two UCBs, where the first UCB allows every agent to additionally use the observations from the other agents to accelerate exploration, while the second UCB uses an NN for reward prediction.  Regret bounds of the proposed algorithms are derived based on NTK techniques and experimental results show the efficacy of the proposed algorithms on both synthetic and real-world data. The reviewers are unanimously positive about the paper. There are shared concerns raised on the novelty of the theoretical analysis as it mainly combines existing analysis technical of federated bandits and neural bandits. However, reviewers agree that as the first study of federated neural bandits, the results are timely and could be interesting to the community. The authors are encouraged to revise the paper following the discussion with reviewers.



**Note From Pc:**

if the above contains the word "oral" or "spotlight" please see: "oral" presentation means -> notable-top-5% and "spotlight" means -> notable-top-25%. As stated in our emails, we are disassociating presentation type from AC recommendations

**Summary Of Ac-Reviewer Meeting:**

Reviewers shared their thoughts on excitement and remaining concerns. While no reviewer is willing to champion the paper, the reviewers are unanimously positive. During the meeting, one remaining concern is raised: theoretical novelty is limited. The proof mainly combines existing analysis technical of federated bandits and NTK based neural bandits. However, reviewers agree that as the first study of federated neural bandits, the results are timely and could be interesting to the community.